# Identification of the transcription factor ZEB1 as a central component of the adipogenic gene regulatory network

Carine Gubelmann[1†], Petra C Schwalie[1†], Sunil K Raghav[1†‡], Eva Röder[2], Tenagne Delessa[2], Elke Kiehlmann[2], Sebastian M Waszak[1§], Andrea Corsinotti[3¶], Gilles Udin[1], Wiebke Holcombe[1], Gottfried Rudofsky[4], Didier Trono[3], Christian Wolfrum[2]*, Bart Deplancke[1]*

[1]Institute of Bioengineering, School of Life Sciences, Ecole Polytechnique Fédérale de Lausanne and Swiss Institute of Bioinformatics, Lausanne, Switzerland; [2]Institute of Food Nutrition and Health, Eidgenössische Technische Hochschule Zürich, Schwerzenbach, Switzerland; [3]Global Health Institute, School of Life Sciences, Ecole Polytechnique Fédérale de Lausanne, Lausanne, Switzerland; [4]Ärztlicher Leiter Endokrinologie, Diabetologie und Klinische Ernährung Kantonsspital Olten, Olten, Switzerland

**\*For correspondence:** christian-wolfrum@ethz.ch (CW); bart.deplancke@epfl.ch (BD)

†These authors contributed equally to this work

**Present address:** ‡Laboratory of Immuno-Genomics and Systems Biology, Institute of Life Sciences, Bhubaneswar, India; §Genome Biology Unit, European Molecular Biology Laboratory, Heidelberg, Germany; ¶MRC Centre for Regenerative Medicine, Institute for Stem Cell Research, School of Biological Sciences, University of Edinburgh, Edinburgh, Scotland

**Competing interests:** The authors declare that no competing interests exist.

**Abstract** Adipose tissue is a key determinant of whole body metabolism and energy homeostasis. Unraveling the regulatory mechanisms underlying adipogenesis is therefore highly relevant from a biomedical perspective. Our current understanding of fat cell differentiation is centered on the transcriptional cascades driven by the C/EBP protein family and the master regulator PPARγ. To elucidate further components of the adipogenic gene regulatory network, we performed a large-scale transcription factor (TF) screen overexpressing 734 TFs in mouse pre-adipocytes and probed their effect on differentiation. We identified 22 novel pro-adipogenic TFs and characterized the top ranking TF, ZEB1, as being essential for adipogenesis both in vitro and in vivo. Moreover, its expression levels correlate with fat cell differentiation potential in humans. Genomic profiling further revealed that this TF directly targets and controls the expression of most early and late adipogenic regulators, identifying ZEB1 as a central transcriptional component of fat cell differentiation.

## Introduction

Obesity and its associated diseases such as diabetes and hypertension affect a high percentage of the world population (*von Ruesten et al., 2011*). Excess fat mass in obese individuals is driven by an increase in adipocyte cell size (hypertrophy) or number (hyperplasia) (*Stephens, 2012*; *Rosen and Spiegelman, 2014*). A comprehensive knowledge of adipocyte differentiation is therefore relevant and timely both from a basic and medical research perspective. Adipocytes arise from multipotent mesenchymal stem cells (MSCs) through an initial lineage commitment phase, followed by terminal differentiation with accumulation of lipid droplets (*Cawthorn et al., 2012*). Although many of the key transcription factors (TFs) controlling this adipogenic program are known, a cohesive model of the underlying molecular events has yet to be revealed.

Most of our current understanding of adipogenesis has been derived using the in vitro murine committed pre-adipocyte cell line 3T3-L1 (*Green and Kehinde, 1975*, *1976*), which to a certain extent recapitulates key features of adipogenesis (*Green and Kehinde, 1979*). The adipocyte differentiation program is accomplished by the expression and activity of a cascade of TFs (reviewed in [*Rosen and Spiegelman, 2000*; *Rosen et al., 2000*; *Farmer, 2006*; *Rosen and MacDougald, 2006*; *Siersbæk and Mandrup, 2011*]), most notably the early-expressed CCAAT/enhancer binding proteins beta (C/EBPβ)

**eLife digest** The growing rates of obesity and related metabolic diseases are a major public health concern worldwide. People who are overweight or obese are at increased risk for a range of diseases including diabetes and heart disease, which may reduce their quality of life and shorten their lifespans. Obese people have more, larger fat cells than individuals of healthy weight, and so understanding how the body creates fat cells may provide new insights into ways to prevent or treat obesity.

Fat cells arise from a population of stem cells that can also give rise to bone cells and cartilage. Some of the proteins—called transcription factors—that work together to turn on the expression of genes needed for a stem cell to become a fat cell have been identified. However, the exact regulatory processes that cause an unspecialized cell to develop into a fat cell remain unclear.

Gubelmann et al. set out to identify more of the transcription factors that cause stem cells to become fat cells. A high-throughput, automated process was used to test the effect of over-expressing each of 734 transcription factors in mouse cells that are primed to become fat cells. Twenty-six transcription factors were found to increase the number of these primed cells that became mature fat cells—most of which had not previously been shown to affect how fat cells develop. The most powerful driver of fat cell development was ZEB1: a transcription factor that has previously been implicated in many other biological processes. Most notably, ZEB1 was linked to a transition during development that allows cells to migrate to the correct location in the embryo, but also to a mechanism that allows cancerous cells to spread to new tissues.

Using studies of mouse cells and live mice, computational analyses, and biopsies from obese patients, Gubelmann et al. show that ZEB1 regulates numerous other transcription factors that promote the development of fat cells. These include factors that initially set an unspecialized cell on the path to becoming a fat cell and those that guide the changes that occur as the fat cell matures. Further studies will be required to show whether the ZEB1 protein itself is needed to prime stem cells to start becoming fat cells.

and delta (C/EBPδ), which induce C/EBPα and the adipogenic master regulator nuclear hormone receptor peroxisome proliferator-activated receptor gamma (PPARγ). C/EBPα and PPARγ cooperate to activate adipogenic genes and cross-regulate each other in a positive feedback loop to maintain the terminally differentiated state of mature adipocytes (*Wu et al., 1999*).

Several studies have revealed the importance of other TFs in regulating adipogenesis both in vitro and in vivo (*Soukas et al., 2001*). These include members of the Krüppel-like family (KLF4 [*Birsoy et al., 2008*], KLF5 [*Oishi et al., 2005*], and KLF15 [*Mori et al., 2005*]), ADD1/SREBP1c (*Fajas et al., 1997*; *Kim et al., 1998*), the E2F (*Fajas et al., 2002*), and the interferon regulatory factor (IRF) families (*Eguchi et al., 2008*), STAT5A/B (*Floyd and Stephens, 2003*) and GATA2/3 (*Tong et al., 2000*). Efforts are therefore underway to integrate this information into comprehensive adipogenic regulatory networks (aGRNs) (*Rosen and MacDougald, 2006*, *Siersbæk et al., 2012*), to which new nodes such as the early regulators ZFP432 (*Gupta et al., 2010*), TCF7L1 (*Cristancho et al., 2011*), or EVI1 (*Ishibashi et al., 2012*) keep being added. This suggests that many important components of this aGRN remain to be discovered.

To systematically identify novel aGRN members, we performed a large-scale overexpression screen with 734 full-length mouse TFs in 3T3-L1 cells, which led to the identification of 26 pro-adipogenic TFs. We found that the top ranking TF, ZEB1, previously known for its role in epithelial to mesenchymal transition (EMT) and tumor metastasis (*Vandewalle et al., 2009*; *Gheldof et al., 2012*), is a critical mediator of in vitro and in vivo adipogenesis, as it directly controls the majority of aGRN genes.

## Results

### A large-scale TF overexpression screen identifies novel positive regulators of adipogenesis

To systematically identify TFs enhancing adipogenesis in an unbiased manner, we tested the effect of overexpressing almost half (48%) of all predicted mouse TFs on 3T3-L1 fat cell differentiation. We transferred 750 available mouse TF open reading frames (ORFs) (*Gubelmann et al., 2013*) into Tet-On Gateway-compatible inducible lentiviral expression vectors and obtained 734 clones (*Figure 1—figure*

*supplement 1A*). Using a robotic platform, lentiviral particles containing each TF ORF were produced and were then used to transduce 3T3-L1 cells with three replicates to ensure reproducibility. We over-expressed each of the 734 TFs in 3T3-L1 cells at confluence and during terminal adipocyte differentiation ('Materials and methods'; *Figure 1A* and *Figure 1—figure supplement 1A*). After 7 days, we stained cells for lipids with the lipophilic, fluorescent dye BODIPY, nuclei with Hoechst, and complete cells with

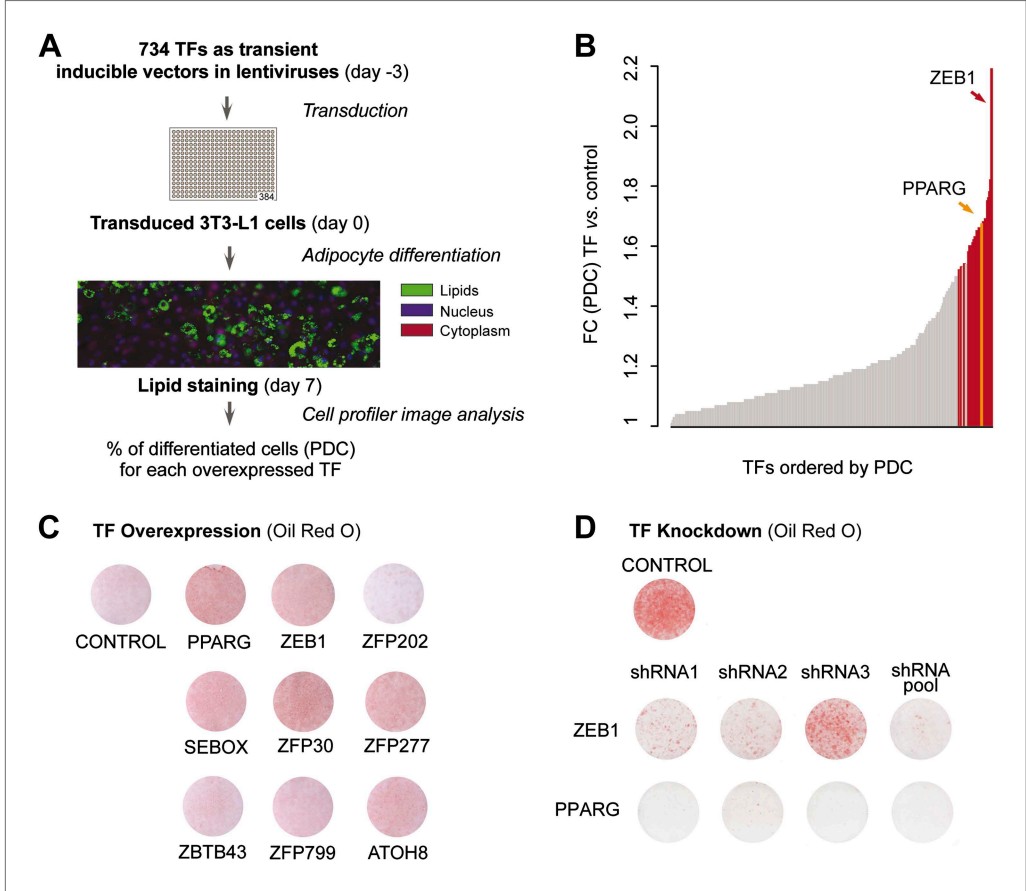

**Figure 1**. A large-scale TF overexpression screen identifies novel positive regulators of adipogenesis. (**A**) Schematic overview of high-throughput screening illustrating how 3T3-L1 cells were transduced with 734 individual TFs in three replicates each, 3 days before induction of adipocyte differentiation ('Materials and methods'). The effect of TF overexpression was quantified at differentiation day 7 by lipid, nucleus and cellular staining and summarized as a percentage of differentiated cells (PDC) per TF. (**B**) Overview of fold-changes (FC) compared to control for all TFs showing a differentiation FC > 1. TFs that significantly induced differentiation (FC ≥ 1.5, α = 0.05) are highlighted in red and PPARγ specifically in orange. (**C**) Effect of stably overexpressing eight putatively novel regulators of adipogenesis, PPARγ, or a control vector on 3T3-L1 differentiation as assessed by Oil Red O staining of lipid droplets at day 5 after induction. (**D**) Effect of knocking down ZEB1 or PPARγ (as a positive control), or the negative control (empty shRNA) on 3T3-L1 differentiation as assessed by Oil Red O staining at day 6 after induction. In the shRNA pool of ZEB1, shRNA2 was not used because the robustness of the cells after treatment was low. Examples of microscopic images illustrating the overexpression or knockdown (KD) effects on 3T3-L1 differentiation are shown in *Figure 1—figure supplement 2* or *Figure 1—figure supplement 3*, respectively.

The following figure supplements are available for figure 1:

**Figure supplement 1**. Large-scale TF overexpression screen identifies novel positive regulators of adipogenesis.

**Figure supplement 2**. Microscopic images of Oil Red O stained 3T3-L1 adipocytes after overexpression of candidate TFs.

**Figure supplement 3**. Microscopic images of Oil Red O stained 3T3-L1 adipocytes after ZEB1 and PPARG KD using distinct KD constructs.

SYTO60 and calculated the percentage of mature adipocytes per total number of cells in each well using automated image analysis as described previously (*Meissburger et al., 2011*) (*Figure 1A* and 'Materials and methods'). Using this approach, we were able to define the percentage of cells that had undergone differentiation (percentage of differentiated cells, PDC), which was used to evaluate the impact of each tested TF on adipogenesis.

Specifically, we compared the PDC obtained for each TF to that of the negative control (original lentiviral vector). We found 26 TFs that significantly enhance adipogenesis (≥ 1.5-fold relative to control (FC), Bonferroni < 0.05) of which 22 have to our knowledge never been implicated in this process, with the TF ZEB1 showing the strongest effects (*Supplementary file 1A* [*Gubelmann et al., 2014*] and *Figure 1B*). Importantly, the master regulator PPARγ was among these 26 stringently selected top ranking candidates, serving as a positive control.

Because our screen was blind to the expression and functional characteristics of the overexpressed TFs, we investigated which of our top enhancing TFs are also endogenously expressed in 3T3-L1 cells or other pre-adipocyte cell lines. For this purpose, we analyzed publicly available microarray expression data from mouse 3T3-L1 and primary human adipose stromal cells, as well as RNA polymerase II (POLII) binding data for expressed TFs (*Nielsen et al., 2008*; *Mikkelsen et al., 2010*). We found that 18 of the top 26 candidates (69%) show high microarray signal or POLII gene body occupancy, suggesting that they are actively transcribed during 3T3-L1 differentiation (*Figure 1—figure supplement 1B*, *Supplementary file 1A* [*Gubelmann et al., 2014*] and 'Materials and methods'). Additionally, we found using Expression Atlas (*Kapushesky et al., 2012*) that nine of the positive candidates (including PPARγ and the top candidate ZEB1) are expressed significantly higher in adipose tissue compared to their mean expression across tissues (*Figure 1—figure supplement 1B*). The high abundance of these TFs may point to a regulatory function in adipocyte-specific processes. Moreover, 15 (83%) of the candidates' human orthologs are also highly expressed in human adipose stromal cells (*Figure 1—figure supplement 1B*), suggesting that their regulation is conserved.

To validate the adipogenic activity of our newly identified pro-adipogenic TFs, we generated 10 stably transduced 3T3-L1 cell lines, including for eight of the top candidates that were not previously implicated in adipogenesis (referred to as 'follow-up TFs'), as well as for PPARγ and the negative control vector, using puromycin selection. TFs were overexpressed for 2 days before inducing differentiation, and lipids were stained with Oil Red O 5 days after differentiation. Overexpression of the TFs was confirmed by Western blots (*Figure 1—figure supplement 1C*). Compared with cells transduced with the control vector, overexpression of 6 out of these 8 TFs led to increased lipid content within the respective cells (*Figure 1C* and *Figure 1—figure supplement 2*) and a strong induction of adipogenic gene expression as measured by quantitative real-time polymerase chain reaction (qPCR) (*Pparg2*, *Cebpa*, and *Adipoq*) after differentiation (*Figure 1—figure supplement 1D–F*). Interestingly, all but one of these six follow-up TFs were expressed at their maximal level in wild-type 3T3-L1 cells already at the onset of adipogenesis (day −2 or day 0), in contrast to *Pparg*, which is highly induced upon differentiation (*Tontonoz et al., 1994*) (*Figure 1—figure supplement 1G* and *Supplementary file 1A* [*Gubelmann et al., 2014*]). This suggests that these TFs may be important in defining the pre-adipocyte state and may act as early regulators of fat cell differentiation.

To confirm the involvement of the top differentiation-enhancing TF ZEB1 in 3T3-L1 adipogenesis, we reduced its expression levels with three distinct shRNAs. In parallel, shRNAs targeting *Pparg* were used as a positive control. After transducing each shRNA into 3T3-L1 cells, we induced differentiation and stained for lipid accumulation at day 6. The pooled knockdown (KD) reduced *Zeb1* gene expression during adipogenesis by approximately 80–90% (*Figure 1—figure supplement 1H*). Oil Red O staining revealed a dramatic reduction in differentiation for individual shRNAs and an almost completely abolished differentiation when the shRNA pool was used, an effect that mimicked that of PPARγ KD (*Figure 1D* and *Figure 1—figure supplement 3*).

Thus, we identified several TFs that increase adipogenesis when transiently or stably overexpressed in 3T3-L1 cells (*Figure 1B–C*). In addition, we revealed that KD of the top adipogenic candidate ZEB1 inhibits adipogenesis in 3T3-L1 cells (*Figure 1D*), suggesting that this TF is a so far unrecognized, important mediator of 3T3-L1 fat cell differentiation.

## Alteration of nuclear ZEB1 levels perturbs the expression of adipogenic regulators and pathways

To explore the mechanism underlying ZEB1-induced stimulation of adipogenesis, we used 3T3-L1 cells. First, we quantified its expression level by qPCR at six adipogenic differentiation time points

(*Figure 2—figure supplement 1A*). Unlike *Pparγ*, whose expression is highly induced upon adipogenesis (*Tontonoz et al., 1994*), *Zeb1* mRNA levels were already high in pre-adipocytes and moderately but significantly decreased during terminal differentiation (*Figure 2—figure supplement 1A–B*, p = 0.009, Wilcoxon rank-sum test days −2 to 2 vs. day 4). This result is consistent with data from previously published microarray-based gene expression during adipogenesis (*Mikkelsen et al., 2010*) as well as with data comparing pre-adipocyte to adipocyte gene expression (Expression Atlas: [*Kapushesky et al., 2012*]) (*Figure 1—figure supplement 1G* and *Figure 2—figure supplement 1C*). ZEB1 may thus already be active at early stages of adipogenesis, in line with the observation that it is among several genes that were highly upregulated immediately after adipogenic induction of mouse embryonic stem cells (*Billon et al., 2010*).

We next examined ZEB1 protein levels during differentiation using our recently developed quantitative proteomics assay (*Simicevic et al., 2013*). We found that ZEB1 is expressed at comparable levels to the nuclear receptor RXRα at day 0 (about 0.25 fmol/µg nuclear extract) ([*Simicevic et al., 2013*] and *Figure 2A*). We observed a ZEB1 protein increase of about 1.4- to 2.1-fold at day 2 compared to day 0 after which ZEB1 decreased to intermediate levels (*Figure 2A* and *Figure 2—figure supplement 1D*). These results indicate that, even though ZEB1 is already highly expressed in pre-adipocytes, its nuclear protein level tends to further increase over the course of differentiation, which appears consistent with the enhancing effect of ZEB1 upon overexpression. This effect may be explained through post-transcriptional regulation.

We next assessed whether the expression of key adipogenic transcriptional regulators is sensitive to nuclear ZEB1 levels. Indeed, ZEB1 overexpression increases *Pparg2* and *Cebpa* levels already in pre-adipocytes and later after induction of differentiation at day 4 (*Figure 2B* and *Figure 2—figure supplement 1E*). Conversely, reducing ZEB1 levels prevents *Pparg2* and *Cebpa* induction as measured at day 4, and significantly reduces their expression in pre-adipocytes (*Figure 2B* and *Figure 2—figure supplement 1E*). To gain global insights into gene expression alterations upon ZEB1 KD, we performed replicate RNA-seq experiments in control and ZEB1 KD cells prior to differentiation (day 0) and 2 days after the onset of differentiation ('Materials and methods'). As expected, *Zeb1* mRNA levels were significantly reduced in both data sets (*Figure 2C*, FC cutoff 1.5 and *padj* ≤ 0.01). Further, the expression fold-changes of several adipogenic TFs and markers measured by qPCR and RNA-seq were highly correlated (Pearson's r ≥ 0.95; *Figure 2—figure supplement 1F*), validating expression estimates obtained by RNA-seq.

In total, 3,426 (17% of all expressed) and 3,221 (16% of all expressed) genes were significantly de-regulated in ZEB1 KD cells compared to control samples at day 0 and day 2, respectively (*Figure 2C* and *Supplementary file 1B* [*Gubelmann et al., 2014*]). We observed no difference between the fractions of genes that are significantly up- or down-regulated after KD (*Figure 2C*), consistent with a summarizing report indicating that the regulatory function of ZEB1 is versatile and may be context-dependent (*Gheldof et al., 2012*), although indirect regulatory effects cannot be excluded.

Genes down-regulated upon ZEB1 KD as measured at day 2 are enriched for fat-cell-specific pathways such as 'Putative pathways for stimulation of fat cell differentiation by Bisphenol A', 'Role of Diethylhexyl Phthalate and Tributyltin in fat cell differentiation' and 'RXR-dependent regulation of lipid metabolism via PPAR, RAR, and VDR' (*Figure 2C* and *Supplementary file 1C* [*Gubelmann et al., 2014*]). Pathway enrichment analysis of genes whose expression changed already in pre-adipocytes revealed a clear distinction between up- and down-regulated genes. While the former enriched for cell cycle and cytoskeleton remodeling-related processes, the latter singled out various developmental pathways previously implicated in adipogenesis (*James, 2013*), including 'TGF-beta-dependent induction of EMT via SMADs', 'Ligand-dependent activation of the ESR1/AP1 pathway', 'WNT signaling pathway Part 2', and 'Notch Signaling Pathway' (*Figure 2C* and *Supplementary file 1C* [*Gubelmann et al., 2014*]). Specifically, reducing ZEB1 expression in 3T3-L1 cells had a strong impact on the expression of the adipogenic master regulators *Cebpa*, *Cebpd* as well as the known adipogenic TFs *Nr3c1* and *Krox20* (*Supplementary file 1B* [*Gubelmann et al., 2014*] and *Figure 2C* and *Figure 2—figure supplement 1F*). Moreover, we found that the expression of four novel adipogenic candidates identified in the TF screen (*Zfp30*, *Ebf3*, *Msx1*, and *Atoh8*) is equally perturbed (at day 0) by reduced ZEB1 levels, suggesting that they may belong to the same regulatory module (*Figure 2C* and *Supplementary file 1B* [*Gubelmann et al., 2014*]).

To better understand the gene regulatory function of ZEB1, we analyzed the proportion of up- and down-regulated genes in each of nine distinct gene clusters that were previously derived from

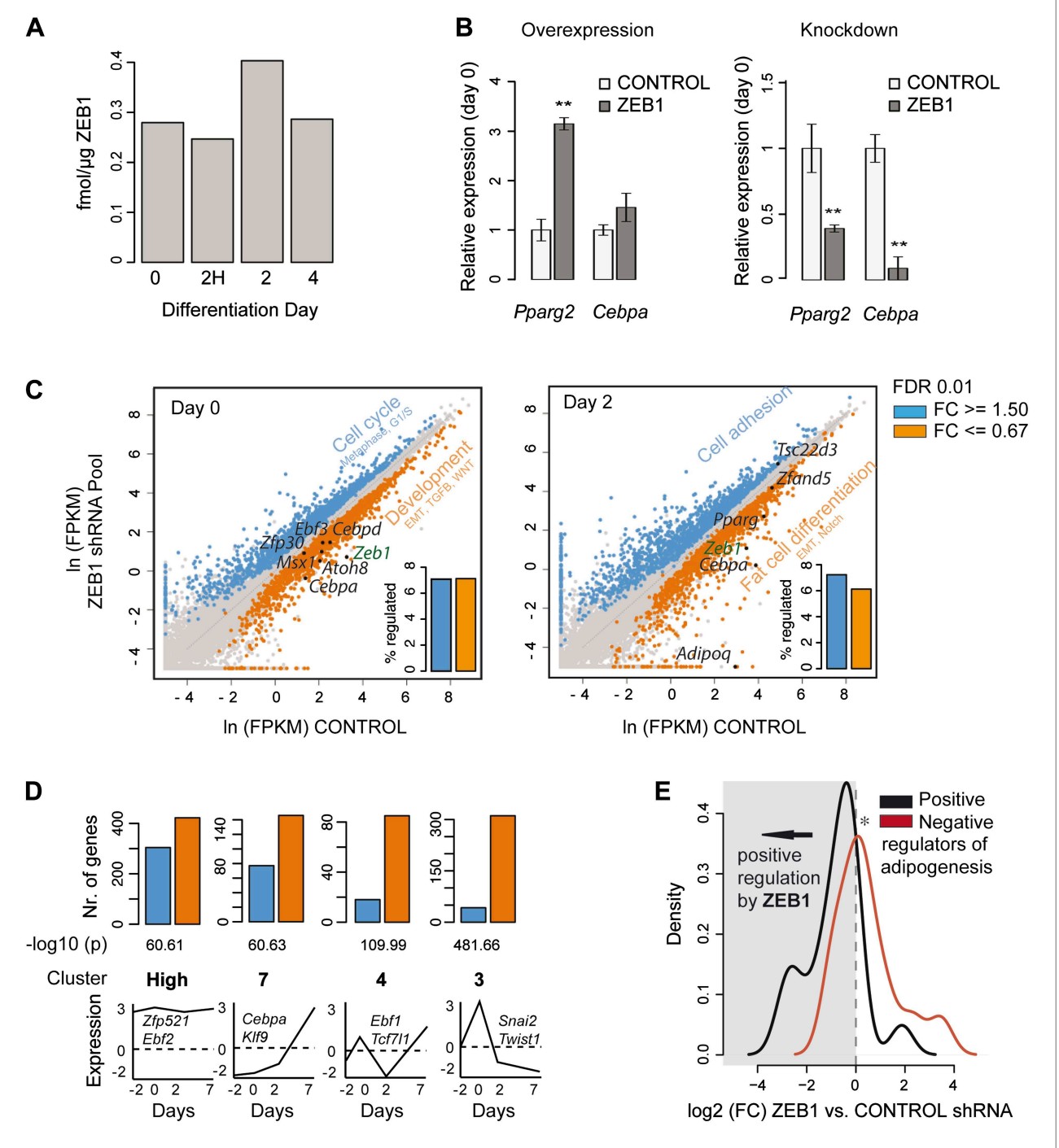

**Figure 2**. ZEB1 knockdown perturbs the expression of adipogenic regulators. (**A**) Protein levels (fmol/µg nuclear extract) of ZEB1 during 3T3-L1 differentiation (one representative biological replicate). (**B**) *Pparg2* and *Cebpa* mRNA levels after ZEB1 knockdown and overexpression in un-induced 3T3-L1 pre-adipocytes as measured by qPCR. (**C**) Expression levels [ln(FPKM), 'Materials and methods'] of mouse genes in ZEB1 KD vs. control cells at day 0 and day 2 after differentiation induction as measured by RNA-seq. Significantly up-regulated genes (FC ≥ 1.5, *padj* ≤ 0.01) are highlighted in blue, down-regulated genes in orange (FC ≤ 0.67, *padj* ≤ 0.01), significantly de-regulated follow-up TFs as well as adipogenic TFs such as PPARγ and C/EBPs are indicated in black. Bar plots represent the percentage of genes that are significantly up- or down-regulated. Representative enriched GeneGO pathway categories for up- or down-regulated genes are highlighted (complete results in ***Supplementary file 1C*** [***Gubelmann et al., 2014***]). (**D**) Number of significantly up- or down-regulated genes belonging to previously defined expression clusters (High/7/4/3) (***Mikkelsen et al., 2010***). The typical expression pattern of genes in each cluster as well as of representative members that are significantly down-regulated upon ZEB1 KD is sketched. *Figure 2. Continued on next page*

*Figure 2. Continued*

Clusters are sorted by increasing enrichment of down-regulated genes and corresponding p-values (chi-square test) are listed. (**E**) Distribution of gene expression FCs at day 0 after ZEB1 KD for genes annotated as positive or negative regulators of adipogenesis (*Supplementary file 1B* [*Gubelmann et al., 2014*]). Error bars depict the standard error of the mean. **$p \leq 0.01$ and $0.01 < *p \leq 0.05$.

The following figure supplement is available for figure 2:

**Figure supplement 1**. ZEB1 knockdown perturbs the expression of adipogenic regulators.

genome-wide microarray-based expression data generated during 3T3-L1 terminal differentiation (*Mikkelsen et al., 2010*). Strikingly, genes showing decreased expression upon confluence under normal conditions (i.e., expression at day −2 is higher than at day 0) were significantly up-regulated at day 0 in ZEB1 KD cells (*Figure 2—figure supplement 1G*). The enrichment was strongest for genes in cluster 2, which were previously associated with mitosis and cell cycle functions. This is consistent with the pathway analysis results reported above and suggests that ZEB1 may control the expression of genes involved in coordinated cell cycle arrest, an essential step in 3T3-L1 differentiation (*Tang et al., 2003*). Conversely, we found that a striking majority of genes whose expression increases upon confluence (day −2 expression lower than day 0 expression) were downregulated at day 0 upon ZEB1 KD (*Figure 2D*). The strongest signal was observed for cluster 3 genes, which were previously associated with cell adhesion and extracellular matrix functionality. We note that several other EMT factors such as *Snai2* (previously shown to be a promoter of adipogenesis [*Pérez-Mancera et al., 2007*]), *Twist1*, *Id2* and *Id3* fall in this cluster and were significantly down-regulated upon ZEB1 KD.

To test how ZEB1 KD specifically impacts the expression of known adipogenic TFs in a statistical threshold-independent manner, we directly compared expression fold-changes of positive and negative regulators (*Supplementary file 1B* [*Gubelmann et al., 2014*]) of fat cell differentiation (*Figure 2E*). In pre-adipocytes, positive regulators tended to be down-regulated upon ZEB1 KD while negative regulators were up-regulated ($p = 0.007$, Wilcoxon rank-sum test comparing fold-changes), suggesting that ZEB1 acts primarily as a positive regulator in the context of the aGRN, consistent with observations from our phenotype-profiling experiments. We recently dissected the role of SMRT (NCOR2), a transcriptional co-repressor, in fat cell differentiation and showed how it maintains genes in a poised state until adipogenesis is induced (*Raghav et al., 2012*). If ZEB1 is, as hypothesized, an activator of early adipogenic genes, which are partially repressed by SMRT, we expect (1) a high overlap between genes regulated by SMRT and ZEB1; and (2) SMRT and ZEB1 KD to have inverse effects on the expression of the genes that they control. Indeed, we found that the fold-changes induced in POLII binding upon SMRT KD and in gene expression upon ZEB1 KD were significantly negatively correlated (Spearman's $\rho = -0.36$, $p < 10^{-16}$) (*Figure 2—figure supplement 1H*). For example, while SMRT KD significantly increased POLII levels at the positive adipogenic regulators *Cebpa* and *Id4* (*Murad et al., 2010*), ZEB1 KD highly reduced their mRNA levels in pre-adipocytes. Overall, we found that over 40% (compared to an expected 14%) of both the genes that lose SMRT binding upon fat cell differentiation and those that experience a POLII binding change upon SMRT KD are significantly ($p < 0.004$ Fisher's exact test and permutation test) de-regulated upon ZEB1 KD (*Figure 2—figure supplement 1I*).

Collectively, these results provide molecular evidence that ZEB1 is required for the adipogenic transcriptional program by controlling mRNA levels of a broad range of genes during adipogenesis. Specifically, ZEB1 promotes the expression of established key adipogenic regulators such as PPARγ and C/EBPα, and balances against pathways that repress and in favor of those that mediate terminal differentiation.

## ZEB1 co-binds adipogenic regulatory regions with established early regulators such as C/EBPβ

To delineate the direct transcriptional effects of ZEB1 from other indirect layers of regulation, we next performed ZEB1 ChIP-seq in 3T3-L1 pre-adipocytes ('Materials and methods'). Replicate experiments including a pull-down with anti-HA in ZEB1-HA overexpressing 3T3-L1 cells showed high ChIP enrichment compared to negative controls as well as high correlation of read counts (Spearman's $\rho \geq 0.83$) (*Figure 3A* and *Figure 3—figure supplement 1A–C*). ZEB1 exhibited widespread DNA binding (27,854 target regions), including at the genomic locus of the adipogenic master regulator *Pparg* (*Figure 3A–B*). We note that ZEB1 preferentially bound gene-proximal locations, with almost 25% of ZEB1 targets

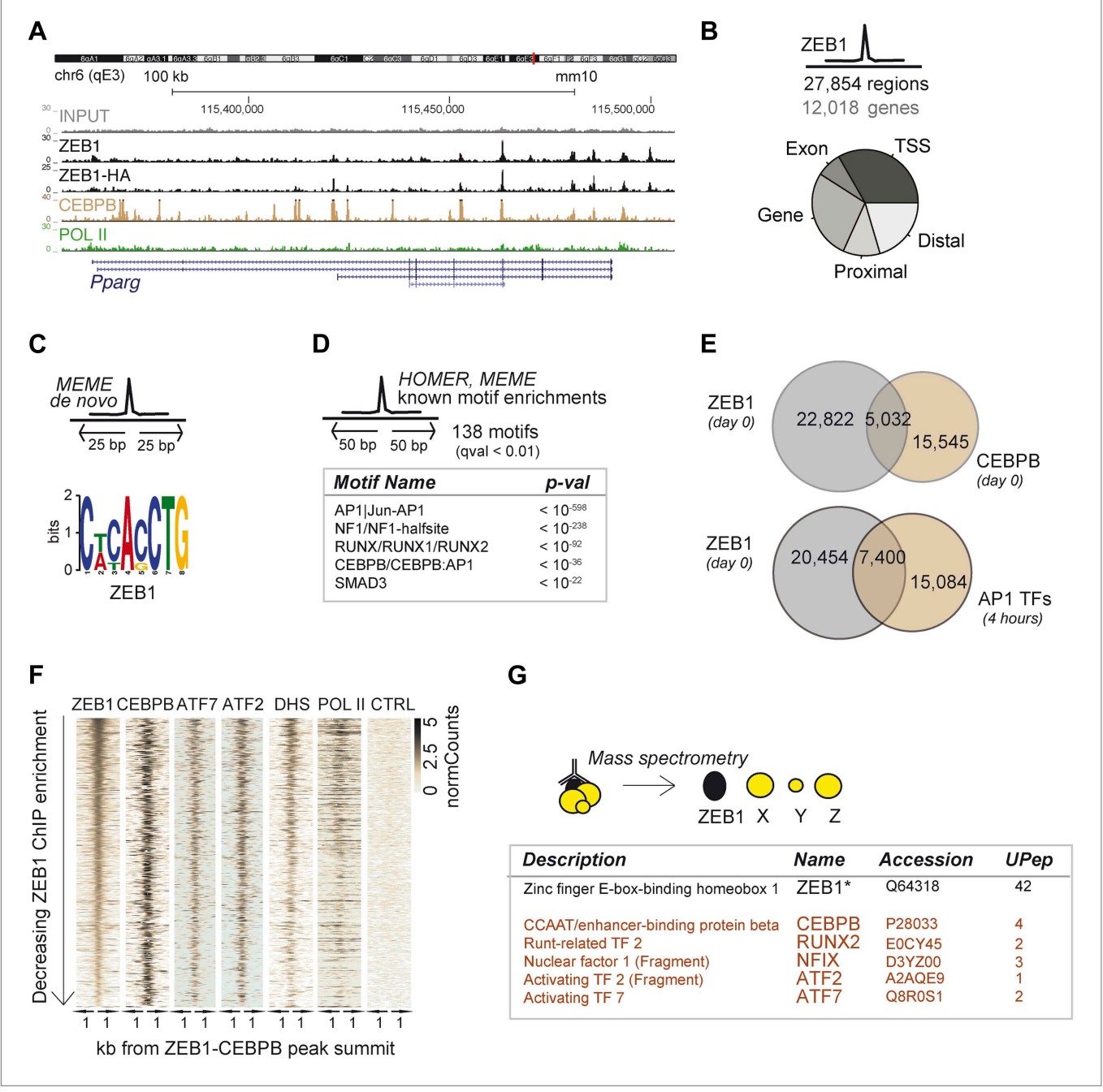

**Figure 3**. ZEB1 co-binds the genome with established adipogenic regulators such as C/EBPβ. (**A**) ZEB1, C/EBPβ, POLII, and Control (CTRL) read density tracks at the *Pparg* locus. (**B**) Number of ZEB1-bound regions and of their proximal (≤ 10 kb) genes in 3T3-L1 cells. Distribution of ZEB1 binding with respect to genomic annotation ('Materials and methods'). (**C**) De novo motif discovery using MEME and a 50 bp sequence centered on ZEB1 peak summits reveals the canonical ZEB1 motif (p = $10^{-8}$, 'Materials and methods') (**D**) Motif enrichment analysis in a 100 bp window around ZEB1 peak summits reveals 138 significantly enriched motifs (complete results in **Supplementary file 1E**). Highlighted here are motif names of the known early adipogenic regulators C/EBPβ, NFI, and AP1 factors as well as RUNX and SMAD3. (**E**) Peak overlap between ZEB1 and C/EBPβ (day 0) as well as AP1 factors (day 0, 4 hr) in 3T3-L1 cells. (**F**) Overview of ZEB1, C/EBPβ, AP1 proteins ATF2 and ATF7, POLII normalized ChIP-seq as well as DNase-seq (DHS) enrichments ('Materials and methods') in a 2 kb window around the summits of ZEB1 peaks that overlap C/EBPβ binding. Intervals are sorted based on decreasing ZEB1 enrichment. (**G**) Summarized results from mass spectrometry experiments of proteins that were identified when pulling down ZEB1 (complete results in **Supplementary file 1F** [**Gubelmann et al., 2014**]).

The following figure supplements are available for figure 3:

*Figure 3. Continued on next page*

*Figure 3. Continued*

**Figure supplement 1**. ZEB1 co-binds the genome with established adipogenic regulators such as C/EBPβ.

**Figure supplement 2**. ZEB1 binding and expression is affected by C/EBPβ KD.

overlapping TSSs and the vast majority of regions located within 10 kb of an annotated gene (*Figure 3B* and *Figure 3—figure supplement 1D* for comparison with random regions, POLII, and C/EBPβ distribution). The association of ZEB1 with promoters as well as with CpG islands and exonic regions is highly significant at a genome-wide scale (p < 10$^{-16}$, *Supplementary file 1D* [*Gubelmann et al., 2014*]).

De novo motif analysis using 50 bp around the ZEB1 peak summit (*Figure 3C*) revealed a top-scoring motif that strongly matched MA0103.2 (TOMTOM p=10$^{-8}$), which was previously annotated as the ZEB1 position weight matrix (PWM) in Jaspar (*Gupta et al., 2007*) (*Figure 3C*). Although the ZEB1 motif was preferentially located proximal to the point of highest ZEB1 ChIP enrichment (*Figure 3—figure supplement 1E*), only a subset (24%) of ZEB1-bound regions contained such a motif match (*Figure 3—figure supplement 1F*). To discover additional enriched sequence patterns of ZEB1-bound regions, we extended the motif search to 100 bp around the peak summit. We obtained a large list of significantly enriched motifs (*Supplementary file 1E* [*Gubelmann et al., 2014*]), including AP1|Jun-AP1, NF1, RUNX, C/EBPβ, and SMAD3 motifs (p ≤ 10$^{-12}$) (*Figure 3D* and *Figure 3—figure supplement 1F*), of which several occurred multiple times within the same ZEB1-bound region (*Figure 3—figure supplement 1F*). Together, these results suggest that, while there is sequence specificity to ZEB1 binding, the protein is most likely recruited to a broad number of regulatory regions, either by or in cooperation with other DNA binding proteins.

To verify whether ZEB1 targets previously identified (pre)adipogenic regulatory regions, we used publicly available C/EBPβ and AP1 binding data in 3T3-L1 cells (*Siersbæk et al., 2011*; *Siersbæk et al., 2014*). We found that almost one third of ZEB1-bound regions were co-bound by C/EBPβ, which represents a significant enrichment over random expectation (Fisher's exact test, p ≤ 10$^{-12}$) (*Figure 3E*). Both ZEB1 and C/EBPβ ChIP-seq signal intensities were high at these 5,032 genomic locations, which were also enriched for AP1 factor binding (in particular ATF7 and ATF2), high DNase I signal, and to a lesser extent POLII (*Figure 3F*). Strikingly, 27% of regions bound by ZEB1 at day 0 were also bound by at least one of the five AP1 complex proteins JUNB, FOSL2, CJUN, ATF7, and ATF2 4 hr after induction of differentiation, again a highly significant co-localization (Fisher's exact test, p ≤ 10–12) (*Figure 3E* and *Figure 3—figure supplement 1G*). Globally, regions showing higher ZEB1 enrichment co-localized more with TF binding and with genomic marks of transcriptional activity such as POLII and H3K9AC as well as open chromatic regions as indicated by high DNase I signal intensity (*Figure 3—figure supplement 1H*). Indeed, genome-wide correlations between ZEB1 and POLII ChIP-seq read counts were high (*Figure 3—figure supplement 1C*). Finally, we also analyzed the overlap between ZEB1, AP1, and C/EBPβ in the human HepG2 as well as lymphoblastoid cell lines (LCLs) (data from ENCODE [*Gerstein et al., 2012*]). We observed consistent enrichment of C/EBPβ, JUN, and FOSL (AP1 complex members) around ZEB1 peak summits (*Figure 3—figure supplement 1I*), providing support for a conserved molecular relationship between these TFs in mediating DNA binding.

To test whether the observed genomic co-localization of ZEB1, C/EBPβ, and AP1 factors relies on physical interactions, we assayed ZEB1 interaction partners in 3T3-L1 cells by mass spectrometry (*Figure 3G* and *Supplementary file 1F* [*Gubelmann et al., 2014*]). We first verified that we could recover the ZEB1 protein as well as its known interaction partners such as CtBP1/2 and HDAC1/2 (*Gheldof et al., 2012*). The presence of these four known partners among the 89 stringently selected ('Materials and methods') and reproducibly pulled down proteins confirmed the validity of our approach. Importantly, we also detected C/EBPβ at similar stringency and additionally, RUNX2, NFIX, and AP1-family members ATF2 and ATF7-specific peptides in one of the replicate experiments (*Figure 3G*). These data suggest that, in 3T3-L1 cells, ZEB1 is located within the same protein complex as at least one of the major regulators of adipogenic gene expression (C/EBPβ) and potentially also cooperates with other adipogenic TFs such as ATF2 or ATF7 to mediate transcription.

To further assess the functional relationship between ZEB1 and C/EBPβ binding, we performed ZEB1 ChIP experiments after stable C/EBPβ knockdown in 3T3-L1 cells (*Figure 3—figure supplement 2A*),

testing 10 ZEB1-C/EBPβ co-bound regions, six ZEB1-only regions, and six negative control regions (*Supplementary file 1I* [*Gubelmann et al., 2014*]) based on our ZEB1 and publicly available C/EBPβ ChIP-seq data (*Siersbæk et al., 2011*). We first validated these C/EBPβ-bound regions by performing C/EBPβ ChIP-qPCR in our (control) 3T3-L1 cells (*Figure 3—figure supplement 2B*). We found that C/EBPβ was almost invariably enriched at ZEB1-bound regions, even at regions (five out of six) that did not show C/EBPβ enrichment in the publicly available ChIP-seq data and that were thus presumed to be bound by ZEB1 only (*Figure 3—figure supplement 2B*). Thus, the genomic co-localization of the two proteins may be even more widespread than originally appreciated, strengthening their functional relationship. In stable C/EBPβ knockdown cells, we observed a decrease of both C/EBPβ and ZEB1 DNA binding at the majority of the tested regions with the exception of the single ZEB1-only bound region, suggesting that ZEB1 DNA-binding is dependent on C/EBPβ (*Figure 3—figure supplement 2C*). However, when we measured *Zeb1* mRNA levels in the C/EBPβ KD cells, we observed a two-fold decrease compared to control (*Figure 3—figure supplement 2D*). This result implies that C/EBPβ mediates *Zeb1* expression, which is substantiated by the fact that C/EBPβ directly targets the *Zeb1* gene in 3T3-L1 pre-adipocytes (*Figure 3—figure supplement 2E*). Thus, at this point, we cannot exclude that the observed decrease in ZEB1 DNA binding in C/EBPβ KD cells may in part be a consequence of decreased cellular ZEB1 levels. Further experiments will be required to examine putative DNA binding cooperativity effects between these two TFs.

Collectively, these observations demonstrate that in committed pre-adipocytes, ZEB1 is bound to open/active regulatory regions already before the onset of adipogenesis. Many of these regions are targeted by first-wave adipogenic TFs such as C/EBPβ and AP1-family members, implying a functional relationship between these TFs and, given the provided evidence, especially between ZEB1 and C/EBPβ in early adipogenic regulatory events.

## ZEB1 DNA binding is dynamic at adipogenic genes

To better understand the dynamic regulatory properties of ZEB1, we profiled its DNA binding during 3T3-L1 fat cell differentiation by including days −2, 2, and 4 in addition to day 0. The vast majority of ZEB1-bound regions (42,050) showed consistent enrichment across all time points (*Figure 4—figure supplement 1A*), reflected in high correlations among samples (Spearman's ρ ≥ 0.75 day 4 vs. any other day; *Figure 4—figure supplement 1B*). However, ZEB1 binding profiles prior to (days −2 and 0, subsequently referred to as 'early') and post (days 2 and 4, referred to as 'late') differentiation induction were more similar to one another, respectively, forming two distinct clusters (*Figure 4—figure supplement 1B*). Globally, we detected 552 early-only and 803 late-only ZEB1-bound regions (FC ≥ 2, FDR 0.1, *Figure 4A–B* and *Figure 4—figure supplement 1A,C*, 'Materials and methods'). Interestingly, genomic loci of several adipogenic regulators, including *Pparg*, *Klf15*, and *Zbtb16* (*Mikkelsen et al., 2010*; *Asada et al., 2011*) contained regions with increased ZEB1 binding after differentiation induction (*Supplementary file 1B* [*Gubelmann et al., 2014*], *Figure 4A* and *Figure 4—figure supplement 1C*).

We next contrasted dynamically and statically ZEB1-bound regions in terms of their sequence properties, enrichment for other factors, as well as types of genes in their proximity. Early-only ZEB1-bound regions enriched for the non-adipogenic motifs RUNX1/2 ($p < 10^{-5}$) and TEAD1 ($p < 10^{-5}$), while late-only binding regions for the sequence motifs C/EBPα/β ($p < 10^{-6}$), NFIC ($p \leq 10^{-3}$), and PPARγ::RXR ($p < 10^{-6}$) ('Materials and methods'; *Figure 4C* and *Figure 4—figure supplement 1D*), indicating that ZEB1 may in part relocate to adipogenic-specific regulatory regions after differentiation induction. Consistently, late-only ZEB1-bound regions showed strong C/EBPβ binding as well as high PPARγ and RXRα enrichments upon differentiation (*Figure 4B*). Moreover, these same regions also featured high DHS and POLII signals at days 2 and 4, respectively, suggesting that they become accessible and transcriptionally active during differentiation in contrast to early-only regions, which showed no to little DNase I hypersensitivity or POLII binding in 3T3-L1 pre-adipocytes (*Figure 4B* and *Figure 4—figure supplement 1A*). As these regions are largely already bound by C/EBPβ in pre-adipocytes (*Figure 4B*), it is possible that ZEB1 specifically relocates there upon differentiation induction to mediate gene activation. Functionally, late-only ZEB1 peaks were associated with genes annotated as important for fat cell differentiation, insulin response, and lipid storage among others (*Supplementary file 1G* [*Gubelmann et al., 2014*], *Figure 4D*), a property distinct from early-only regions (*Supplementary file 1G*, *Figure 4—figure supplement 1E*).

Previously defined patterns of gene expression across 3T3-L1 differentiation (*Mikkelsen et al., 2010*) further support this emerging, functional distinction of early and late ZEB1 enrichment. Specifically, we

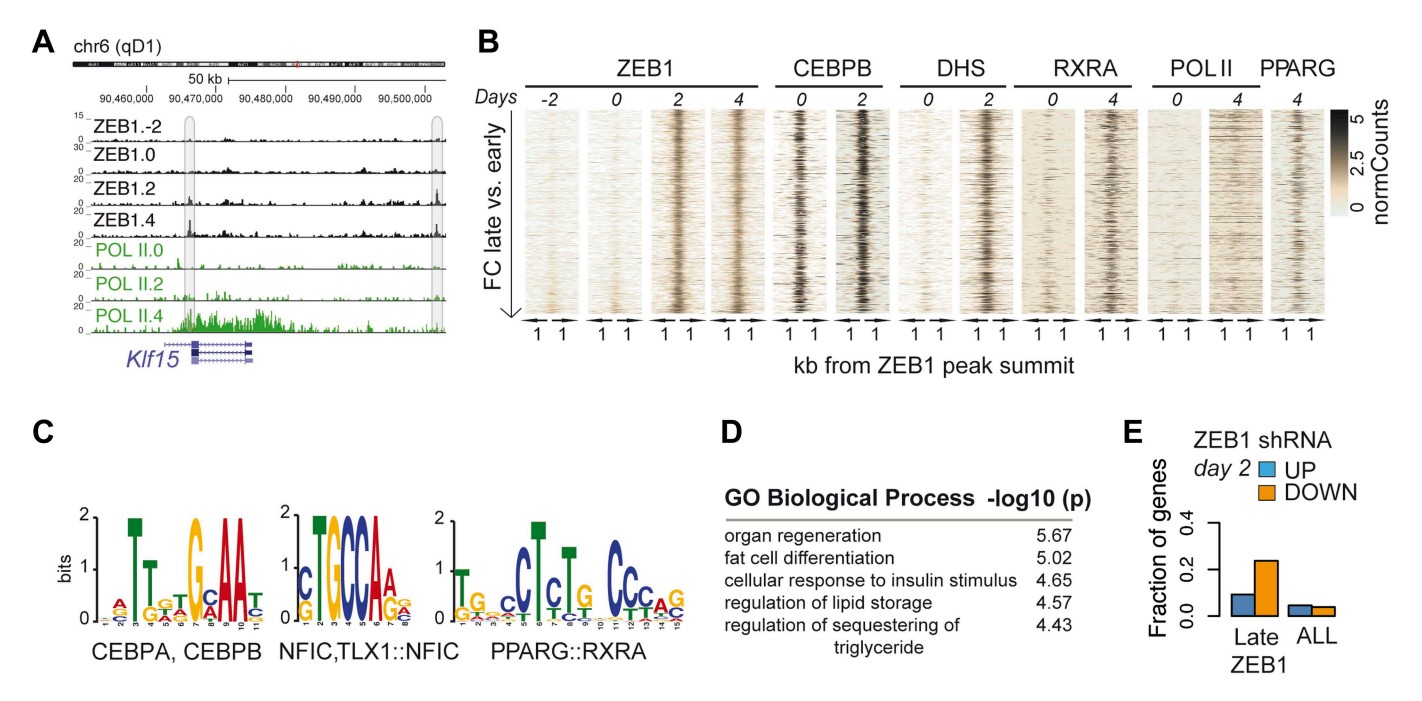

**Figure 4**. ZEB1 binding increases at adipogenic genes during 3T3-L1 differentiation. (**A**) ZEB1 and POLII read density tracks at the *Klf15* locus during 3T3-L1 differentiation (days −2, 0, 2, and 4). Late-only bound regions are highlighted. (**B**) ZEB1, C/EBPβ, RXRα, PPARγ, POLII normalized ChIP ('Materials and methods') as well as DHS enrichments in a 2 kb window around the summits of late-only (days 2 and 4 but not days −2 and 0; *padj* ≤ 0.1, FC ≥ 2) ZEB1-bound regions during 3T3-L1 differentiation. (**C**) Differential motif discovery using MEME and a 50 bp sequence centered on summits of late-only vs. static ZEB1 peaks reveals adipogenic motifs: C/EBPα|C/EBPβ, NFIC and PPARG::RXR (p < 10⁻³, 'Materials and methods'). (**D**) GREAT-based (**McLean et al., 2010**) Gene Ontology enrichment analysis of genes associated with late-only vs. static ZEB1 binding reveals terms associated with fat cell differentiation and function (complete results in **Supplementary file 1G** [**Gubelmann et al., 2014**]). (**E**) Fraction of genes associated with late-only ZEB1 binding and fraction of all genes significantly up (blue) and down (orange)-regulated after ZEB1 KD as measured at differentiation day 2 (complete results in **Figure 4—figure supplement 1G**).

The following figure supplement is available for figure 4:

**Figure supplement 1**. ZEB1 binding increases at adipogenic genes during differentiation.

found that genes marked by early-only ZEB1 binding (≤ 10 kb distance, 'Materials and methods') were much more likely to have low expression in pre-adipocytes or to be repressed upon differentiation (clusters '1' and 'Low') (**Figure 4—figure supplement 1F**). On the other hand, late-only genes enriched for strong induction at either early of late time-points of adipogenesis (expression clusters '7' and '5') (**Figure 4—figure supplement 1F**). To further assess the functional impact of ZEB1 depletion on these late- vs. early-only bound genes, we integrated the expression data into our analyses ('Materials and methods'). The most notable observation was that a large fraction of late-only genes (here defined as genes that contain at least one late-only bound region but no early-only one) are down-regulated upon ZEB1 KD, with almost a quarter of them being significantly lower expressed at differentiation day 2 (**Figure 4E** and **Figure 4—figure supplement 1G**), corresponding to an almost three-fold enrichment compared to statically bound genes (p ≤ 10⁻¹², Fisher's exact test). Collectively, these results are consistent with the observation that an important fraction of genes that are induced during adipogenesis show increased proximal ZEB1 binding after addition of the differentiation cocktail.

We conclude that ZEB1 DNA binding is largely static during adipogenesis and given the strong overlap with DNase I hypersensitive sites may therefore contribute to establishing the regulatory landscape in pre- and mature adipocytes. Nevertheless, a subset of ZEB1 target sites is clearly dynamically bound. Genes that acquire ZEB1 binding after day 0 tend to be also bound by C/EBPβ, up-regulated after differentiation induction, particularly sensitive to alternation of nuclear ZEB1 levels and overall

enriched for adipogenic functions. This suggests that ZEB1 may be involved in mediating the regulatory switch in primed pre-adipocytes toward the terminal differentiation program.

## ZEB1: a core component of the adipogenic GRN

To gain a complete overview of all levels at which ZEB1 regulates adipogenesis, we specifically focused on the aGRN that we manually assembled and curated based on recent reviews as well as Wikipathways (*Rosen and MacDougald, 2006*; *Kelder et al., 2012*; *Siersbæk et al., 2012*). We displayed the

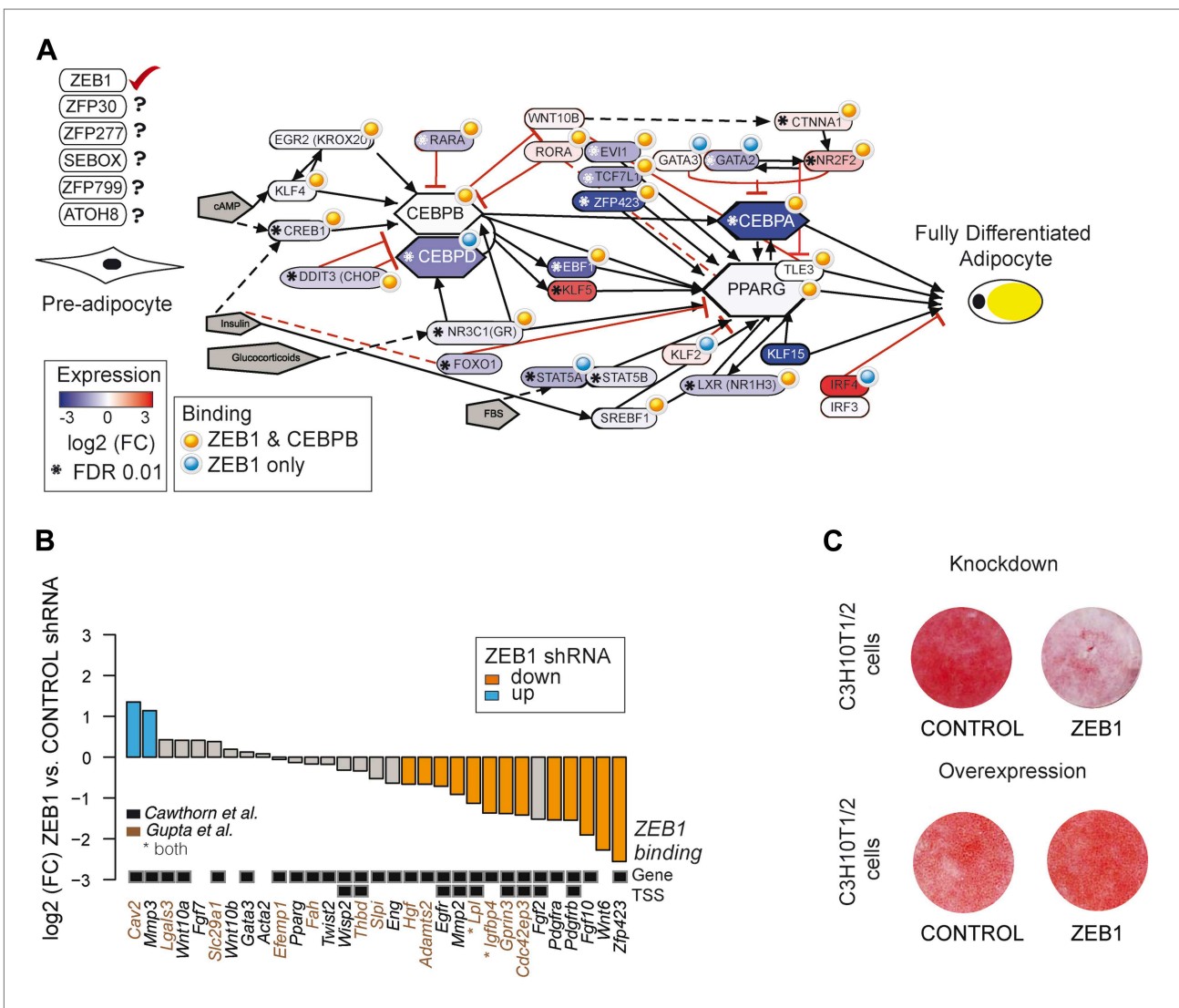

**Figure 5**. ZEB1: a central component of the adipogenic regulatory network. (**A**) Effect of ZEB1 knockdown on the adipogenic gene regulatory network. The network was assembled on the 'Adipogenesis' Pathway scaffold in WikiPathways as well as reviews and most recent publications of novel adipogenic regulators (*Rosen and MacDougald, 2006*; *Kelder et al., 2012*; *Siersbæk et al., 2012*). ZEB1 and C/EBPβ-bound regions that are proximal (within 500 bp) to TSSs or genes in pre-adipocytes are highlighted. *Significant (*padj* ≤ 0.01) expression changes after ZEB1 KD at day 0 of 3T3-L1 differentiation. Other candidate adipogenic regulators identified by our high throughput screen are listed. (**B**) Expression changes of adipogenic commitment genes after ZEB1 KD in 3T3-L1 pre-adipocytes as measured by RNA-seq. Displayed genes are either part of the pre-adipocyte expression signature derived by *Gupta et al.* (2010) or of the list of pre-adipocyte commitment factors compiled by *Cawthorn et al.* (2012). *Lpl* and *Igfbp4* occur in both lists. Significant differences in expression (*padj* ≤ 0.01) are marked in orange (FC ≤ 0.67) and blue (FC ≥ 1.5). Black-grey squares depict ZEB1 binding to TSSs or gene bodies. (**C**) Effect of ZEB1 knockdown and overexpression on C3H10T1/2 adipogenesis as assessed by Oil Red O staining at day 7 and day 8, respectively after induction. DOI: 10.7554/eLife.03346.014

The following figure supplement is available for figure 5:

**Figure supplement 1**. ZEB1 regulates adipogenic commitment factors. DOI: 10.7554/eLife.03346.015

network as well as fold-changes per gene after ZEB1 KD and information on ZEB1 and C/EBPβ binding using Pathvisio (*van Iersel et al., 2008*) (*Figure 5A*). The vast majority of network members are directly targeted by ZEB1 and their expression significantly decreases upon ZEB1 KD (p = 2 × 10⁻⁴ based on permutation, 'Materials and methods'). The number of bound and regulated genes is significantly greater than expected by chance alone, suggesting that ZEB1 is a central component of the aGRN. As expected based on the co-localization of ZEB1 and C/EBPβ, the majority of ZEB1 targets are also targeted by C/EBPβ, suggesting that the two factors cooperate to promote adipogenesis.

One aspect emerging from the regulatory network display is that ZEB1 targets late onset master regulators such as C/EBPα and PPARγ, first wave regulators such as KLF4 and CREB, as well as early adipogenic commitment factors including ZFP423, TCF7L1, and EVI1 (*Figure 5A*). We thus performed a more systematic survey of the extent of ZEB1 binding to genes previously involved in adipogenic commitment, as defined by (1) a specific 3T3 pre-adipocyte expression signature (*Gupta et al., 2010*) and (2) pre-adipocyte factors found to be regulated at the expression level (recently compiled in *Cawthorn et al., 2012*). We found that the expression of about half (48%) of these genes significantly changes after ZEB1 KD in 3T3-L1 cells, with a striking majority (87%) of them being down-regulated. We note that this includes both transcriptional regulators such as the above-mentioned ZFP423, ligands such as WNT6 and membrane receptors such as PDGFRs (*Figure 5B*) and that the majority of their encoding loci are directly targeted by ZEB1 in un-induced pre-adipocytes (*Figure 5B*).

Given ZEB1's involvement in mediating the expression of early adipogenic commitment factors, we next tested whether ZEB1 also regulates adipogenesis in uncommitted precursors such as the C3H10T1/2 MSCs. C3H10T1/2 cells can be committed to the adipocyte lineage by activation of the TGFβ pathway by BMP2/4 and these cells subsequently differentiate into mature adipocytes by the addition of the same hormonal cocktail as used for 3T3-L1 differentiation (*Pinney and Emerson, 1989*; *Tang et al., 2004*). We found that upon shRNA-mediated reduction of ZEB1 levels, the differentiation of MSCs into adipocytes was significantly impaired despite addition of the commitment factor BMP-2 (*Figure 5C*). In addition, central adipogenic TFs and markers, including *Cebpa*, *Pparg2*, *Ebf1*, and *Adipoq*, were significantly down-regulated upon ZEB1 KD (*Figure 5—figure supplement 1A*). We also investigated the effect of ZEB1 overexpression in C3H10T1/2 cells, and observed a slight increase in differentiation potential compared to control cells (*Figure 5C*). In addition, molecular analyses revealed a strong up-regulation of *Pparg2* and *Zfp423* mRNA levels (*Figure 5—figure supplement 1B*) as well as PPARγ protein levels (*Figure 5—figure supplement 1C*), providing further evidence as to the importance of ZEB1 in controlling the expression of this adipogenic master regulator. Collectively, these results indicate that ZEB1 is important for adipogenesis of both 3T3-L1 cells and MSCs.

## ZEB1 is an important regulator of in vivo adipogenesis

Our findings using mouse pre-adipocyte and mesenchymal cell cultures support an active, regulatory role for ZEB1 throughout adipogenesis and speak for a context- and dose-dependent function of this versatile transcriptional regulator. To assess whether these results also translate to an in vivo context, we next investigated the impact of altered ZEB1 levels onto in vivo adipogenesis.

Since *Zeb1* knockout mice are not viable (*Higashi et al., 1997*), we used a previously described method (*Kawaguchi et al., 1998*; *Meissburger et al., 2011*) to probe the effect of ZEB1 overexpression or KD on in vivo adipose stromal-vascular fraction (SVF) cell differentiation. Specifically, GFP-expressing, Matrigel-embedded murine SVF cells that were transduced with lentiviral particles containing either shRNAs or constitutive overexpression constructs for ZEB1 or negative controls were transplanted into the subcutaneous layer of the mouse neck. Mice with transplants were then subjected to a high fat diet for 6 weeks after which the Matrigel pads were analyzed for adipocyte number (*Meissburger et al., 2011*) (*Figure 6A* and *Figure 6—figure supplement 1A*).

Implanted SVF cells serving as controls produced a similar percentage of mature fat cells (~12%), demonstrating high assay reproducibility. SVF cells overexpressing ZEB1 yielded a significantly (p = 0.05, Wilcoxon rank-sum test) higher number of mature fat cells compared to control (*Figure 6B*). Conversely, ZEB1 KD almost halved the formation of mature fat cells in the transplanted pad compared to the negative control (*Figure 6B*). Thus, ZEB1 is also highly important for adipogenesis in vivo. To assess whether the observed ZEB1 overexpression or KD effects could be linked to proliferation changes of adipocyte precursor cells, we quantified the nuclei from the implant sections ('Materials and methods'). We did not observe any change in proliferation in the ZEB1 KD and overexpression samples compared to the respective controls, although ZEB1 KD tended to yield more variable data

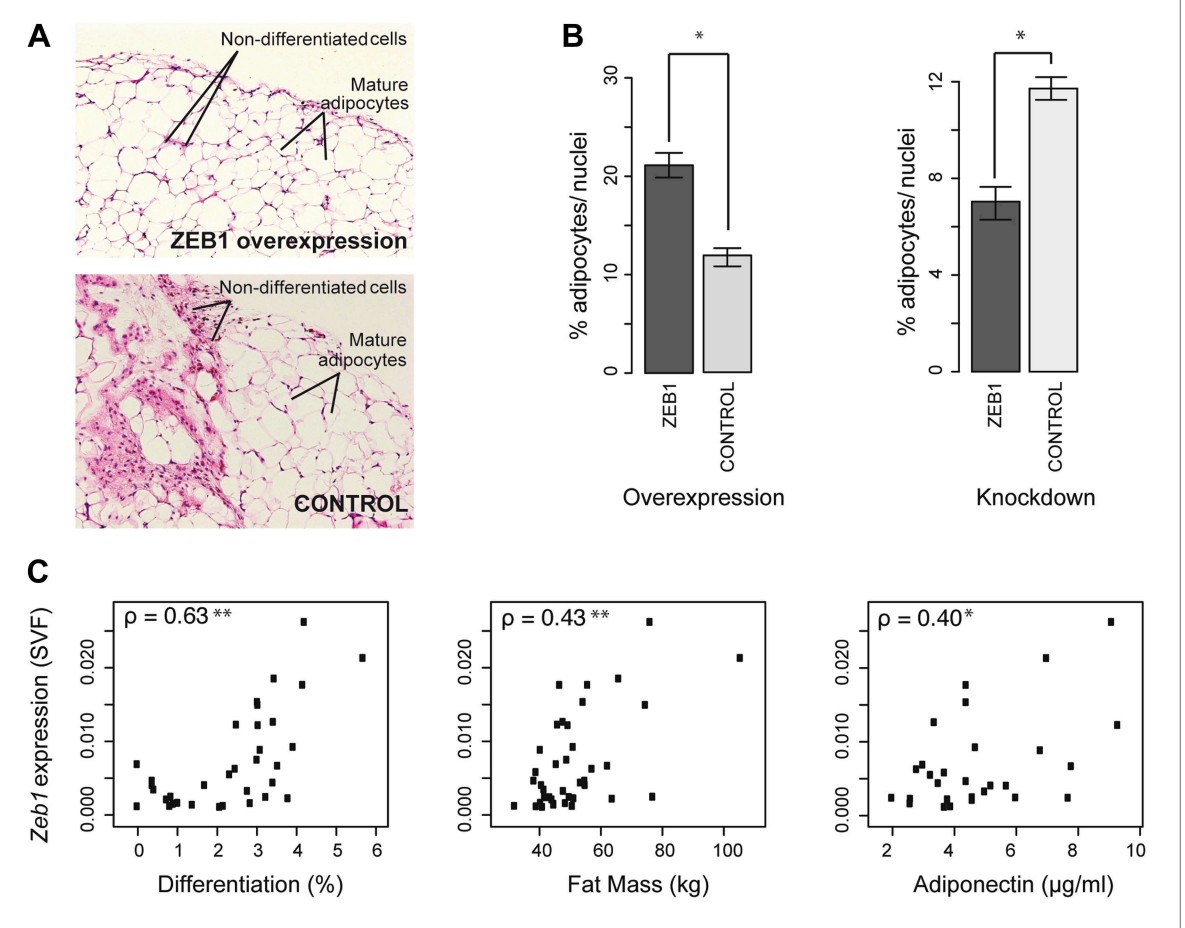

**Figure 6**. ZEB1 is required for adipogenesis in vivo in mouse and its expression levels correlate with adipogenic indicators in humans. (**A** and **B**) Adipocyte differentiation in stromal vascular fraction (SVF) transplants from different donor mice (as indicated) fed a high-fat diet for 6 weeks (*Meissburger et al., 2011*). (**A**) Fat sections from representative samples of ZEB1-overexpressing and control SVF transplants stained with Hematoxylin (blue) and Eosin (pink). (**B**) Fat cell content of the transplanted SVF cells containing ZEB1 and control overexpression or knockdown constructs. Error bars depict the standard error of the mean. *p = 0.05, one-sided Wilcoxon-rank sum test. (**C**) *Zeb1* mRNA expression normalized to *36B4* in human subcutaneous SVF of obese subjects plotted against percent ex vivo differentiated adipocytes of human subcutaneous SVF, subject fat mass, and adiponectin levels. Spearman's ρ is indicated, **p ≤ 0.01 and 0.01 < *p ≤ 0.05.

The following figure supplement is available for figure 6:

**Figure supplement 1**. Analysis of the functional involvement of ZEB1 in mouse in vivo adipogenesis.

(*Figure 6—figure supplement 1B*). These results suggest that the observed effect of ZEB1 on adipogenesis does not involve major changes in the extent of cell proliferation capacity, although more in-depth molecular studies will be required to substantiate these findings.

Having demonstrated ZEB1's importance in mouse adipose biology, we asked whether its highly conserved ortholog would exert a similar function in humans. Interestingly, previous research linked sequence variation of the genomic locus in which *Zeb1* is located to body fat distribution and obesity (*Hager et al., 1998*; *Heid et al., 2010*), supporting a possible role for ZEB1 in these processes. To investigate this, we used subcutaneous fat biopsies from a previously sampled cohort of 62 obese patients (41 females and 21 males) with body mass indices (BMI) of 31–64 kg/m² (*Meissburger et al., 2011*; *Winkler et al., 2013*). Specifically, we determined *Zeb1* mRNA expression in the SVF from each patient's fat sample. We then correlated the resulting expression values to multiple adipose-relevant measures (*Supplementary file 1H* [*Gubelmann et al., 2014*], 'Materials and methods'), including the ex vivo differentiation potential of human SVF from subcutaneous adipose tissue biopsies, as previously described (*Meissburger et al., 2011*). Interestingly, we found strong positive correlations (Spearman's

ρ ≥ 0.40, p ≤ 0.03) with adipocyte differentiation, total fat mass, as well as adiponectin levels (*Figure 6C* and *Supplementary file 1H* [*Gubelmann et al., 2014*]). In contrast, it was previously reported that *Rorg* gene expression levels show an inverse correlation with adipocyte differentiation potential, and no significant correlation with either fat mass or adiponectin (*Meissburger et al., 2011*) (*Supplementary file 1H* [*Gubelmann et al., 2014*]). Thus, *Zeb1* SVF expression levels appear indicative of differentiation potential, fat mass, and adiponectin levels in humans, consistent with the positive effect of ZEB1 on adipogenesis observed in mice. Together, these results demonstrate the functional relevance of ZEB1 in the context of both mouse and human adipocyte biology.

## Discussion

### A TF overexpression screen identifies novel positive regulators of adipogenesis

A comprehensive understanding of the regulatory mechanisms mediating adipocyte differentiation has great fundamental and medical value. Significant efforts have therefore previously been undertaken to uncover adipogenic regulators. These range from classical phenotype- and 'guilt by association'-driven studies (reviewed in *Tang and Lane, 2012*), over enrichment analyses in adipogenic compared to non-adipogenic clonal lines (*Gupta et al., 2010*; *Zhou et al., 2013*), to chromatin state-based inference of key regulators (*Eguchi et al., 2008*; *Mikkelsen et al., 2010*; *Waki et al., 2011*). Of late, more large-scale studies have started to emerge in both mouse and human systems (*Villanueva et al., 2011*; *Söhle et al., 2012*). Our study differs from these high-throughput screening approaches in that we set out to systematically identify novel transcriptional regulators of adipogenesis through an overexpression screen of TFs in 3T3-L1 pre-adipocytes. This was made possible due to the recent generation of a comprehensive library of mouse TF ORF clones (*Gubelmann et al., 2013*), which allowed us to explore the completeness of the so-far assembled aGRN (*Figure 5A*, [*Tang and Lane, 2012*]).

We were able to identify several TFs having strong reproducible effects on fat cell differentiation. The majority of these have so far either never or only indirectly been linked to adipogenesis. For example, the only available information for the second ranking regulator ZFP30 is that it belongs to the family of KRAB zinc finger proteins, typically repressors with important roles in vertebrate development (*Urrutia, 2003*). Other examples include MSX1 and ATOH8, which have been associated with differentiation processes in mesenchymal lineages but to our knowledge never with adipogenesis (*Cheng et al., 2003*; *Jimenez et al., 2007*; *Rawnsley et al., 2013*). More generally, the identification of several novel adipogenic TFs suggests that, besides the well-characterized TF cascade including the C/EBP family of TFs and the canonical master regulator PPARγ, a substantial number of yet unconnected TFs also contribute to the adipogenic phenotype. This underscores the value of our screen, which provides a comprehensive resource that will accelerate the complete characterization of the core GRN underlying fat cell differentiation.

### ZEB1 emerges as a central component of the aGRN

The TF with the strongest effect on fat cell differentiation in our screen was ZEB1. This protein (also known as δEF1, TCF8, NIL2-a, BZP, AREB6, MEB1, ZFHX1a, and ZFHEP) is a highly conserved and versatile TF that was originally identified as a repressor of the lens-specific δ1-crystalline enhancer in chicken (*Funahashi et al., 1991*). It has subsequently been shown to be involved in a broad range of regulatory processes, including EMT, tumor metastasis, development, and differentiation (reviewed in *Gheldof et al., 2012*), providing a rationale as to why homozygous *Zeb1* null mice are not viable (*Higashi et al., 1997*). Similar to the majority of the other TFs identified in our screen, little is known about the involvement of ZEB1 in fat cell differentiation. One report linked a ZEB1 gain-of-function mutation to increased adiposity (*Kurima et al., 2011*), consistent with our findings; another found that female mice that are heterozygous for a targeted deletion of exon 1 of *Zeb1* show increased adiposity (*Saykally et al., 2009*), thus identifying ZEB1 as a repressor of adipogenesis. However, the latter study also reported that ZEB1 expression in parametrial fat increases as fat accumulates, in line with the results presented here. We therefore suspect that the phenotype observed in the studied ZEB1 haploinsufficient mice may not be related to adipocyte hyperplasia, but rather to hypertrophy, another well-established means of adipose tissue expansion. Thus, our findings consolidate and explain ZEB1's involvement in adipogenesis.

Indeed, (1) the large effects that reduction of ZEB1 levels has on the adipogenic phenotype and on the expression of both early (e.g. including ZFP423, TCF7L1 and EVI1) and late (e.g. PPARγ, C/EBPα, KLF15) members of the aGRN, (2) the fact that many of these TFs are also directly targeted by ZEB1, and (3) the fact that ZEB1 operates at the crossroads of several pathways underlying adipogenesis collectively indicate that this TF is an integral part of the molecular identity of pre-adipocytes. This is further supported by earlier observations revealing that *Zeb1* is expressed at high levels in adipose stem cells in vivo (**Kupershmidt et al., 2010**), that it is significantly expressed higher in pre-adipocytes compared to adipocytes (**Kapushesky et al., 2012**), and that it is one of relatively few genes that is highly up-regulated during transition of mouse embryonic stem cells to adipocytes (**Billon et al., 2010**). Interestingly, we found that many of the newly identified TFs such as ZFP30, MSX1, and ATOH8 are also highly expressed in pre-adipocytes. In addition, they are all direct targets of ZEB1 and their expression is affected by ZEB1 KD, suggesting that they may also have a role in early adipogenesis. The identification of several, putatively novel early adipogenic regulators likely reflects the nature of our experimental set-up, which employs early induction of TF overexpression before the cells undergo terminal differentiation. The ability to detect such early regulators is valuable, because the uncharted gene regulatory territory is especially confined to early processes rather than late ones for adipogenesis (**Gupta et al., 2010**).

To provide a mechanistic understanding of how ZEB1 mediates adipogenesis and to further characterize its regulatory mode of action, we assessed genome-wide gene expression differences after ZEB1 KD at days 0 and 2 of fat cell differentiation. In addition, we performed ZEB1 ChIP-seq at distinct time points during adipogenesis, providing a first glance into the dynamic DNA binding properties of ZEB1 in any biological system. We found that ZEB1 DNA binding is widespread, to an extent similar to that of other essential regulators such as C/EBPβ, PU.1, or MYC (**Heinz et al., 2010**; **Siersbæk et al., 2011**; **Nie et al., 2012**), preferentially promoter-proximal and largely static, consistent with its relatively stable expression profile during terminal fat cell differentiation. In terms of sequence specificity, we found that ZEB1-bound regions were not only enriched for E-boxes (in particular 'CACCTG') as previously described, but also for a large number of DNA binding motifs of established adipogenic regulators, including C/EBPβ, AP1, and NFI factors (**Waki et al., 2011**). Together with its strong overlap with DNase hypersensitive sites, these results suggest that ZEB1 contributes to establishing the regulatory landscape in pre- adipocytes as well as mature adipocytes.

Nevertheless, we found that a subset of ZEB1 target regions is dynamically bound. Genes that acquire ZEB1 binding upon differentiation induction tend to be co-bound by C/EBPβ, transcriptionally up-regulated, and enriched for adipogenic functions. These data suggest that ZEB1 may be involved in mediating the regulatory switch in primed pre-adipocytes toward the terminal differentiation program. Interestingly, a comparable mechanism has been uncovered in vascular smooth muscle cells, where ZEB1 has been shown to interact with SMAD3 and to synergistically activate the transcription of key differentiation genes (**Nishimura et al., 2006**). In the context of adipocytes, we propose that ZEB1 forms a complex with C/EBPβ, as supported by our mass spectrometry data, and that these two TFs cooperate to control adipogenic gene expression. In contrast, we found that early-only bound regions were enriched for motifs of the osteogenic master regulator RUNX2, raising the possibility that ZEB1 may bind to osteogenic regulatory regions (likely in a repressive context), and may thus be involved in the bone-fat switch. This is consistent with ZEB1's previous implication in differentiation of various mesenchymal lineages, including osteogenesis, myogenesis, and chondrogenesis (reviewed in [**Gheldof et al., 2012**]).

Importantly, murine ZEB1 shows strong conservation to its human ortholog and its gene was located within one of the top 16 regions of association in a recently performed meta-analysis of 32 body fat distribution genome-wide association studies (**Heid et al., 2010**), consistent with an earlier report indicating a putative link between genomic variation in the *ZEB1* locus and obesity (**Hager et al., 1998**). Our data from obese patients underscore this importance. Specifically, we found that *ZEB1* levels in the SVF correlate with differentiation potential, total fat mass, and circulating adiponectin levels. It is well accepted that adipocyte hyperplasia in obesity can lead to an increased fat cell number with smaller more insulin sensitive adipocytes (**Tilg and Moschen, 2006**; **Roberts et al., 2009**). The positive correlation of *ZEB1* expression with adiponectin levels, which confers insulin sensitivity in a wide variety of tissues, points towards such a scenario. Our study therefore breaks new ground with respect to investigating the functional significance of this association.

# Materials and methods

## Cell culture and differentiation

3T3-L1 mouse fibroblast cells and C3H10T1/2 mesenchymal stem cells, obtained from ATCC, were cultured in high-glucose Dulbecco's modified Eagle's medium (DMEM, Life Technologies, Carlsbad, CA) supplemented with 10% fetal calf serum (FCS, Bioconcept, Switzerland), and 1x penicillin/streptomycin solution (1x Pen/Strep, Life Technologies) in a 5% $CO_2$ humidified atmosphere at 37°C and maintained at less than 80% confluence before passaging. Differentiation of 3T3-L1 cells was induced by exposing 2-day post-confluent cells (day 0) to DMEM containing 10% FCS supplemented with 1 μM dexamethasone, 0.5 mM 3-isobutyl-1-methylxanthine, and 167 nM insulin (Sigma, Saint-Louis, MO), a medium called MDI. Note that we did not add any rosiglitazone since the potent differentiation effect of this PPAR activator would have severely limited our ability to identify positive regulators of adipogenesis. After 2 days (day 2), cells were washed with Dulbecco's Phosphate-Buffered Saline 1x (PBS, Bioconcept, Switzerland) and were fed with DMEM containing 10% FCS and 167 nM insulin. 2 days later (day 4), the medium were changed to 10% FCS/DMEM. Full differentiation was usually achieved by day 6.

To identify the effect of ZEB1 overexpression or knockdown (KD) on differentiation of C3H10T1/2 cells into fat cells, $5 \times 10^4$ cells were plated per well of six well plates. When the cells were 80% confluent, BMP-2 (50 ng/ml, Life Technologies) supplemented complete DMEM medium was added to the cells for commitment into pre-adipocytes. At confluence, day 0, adipocyte differentiation induction medium (MDI supplemented with 50 ng/ml BMP-2) was added to the cells. After 2 days the cells were washed once with 1x PBS and were fed with DMEM containing 10% FCS, 1x Pen/Strep solution containing 167 nM insulin and 50 ng/ml BMP-2. This was followed by the addition of 500 nM rosiglitazone containing complete media at day 4 to the cells. Rosiglitazone containing medium was changed after 2 days and then fat-containing cells were stained with Oil Red O at day 8.

## TF cloning in overexpression lentiviral vector

The generation of our 750 fully sequence-verified mouse TF ORF clones via Gateway cloning is explained in Gubelmann et al. (*2013*). The 750 TF ORFs were subcloned from the pDONR221 entry clones into a Tet-On Gateway-compatible expression vector (IRES-PURO, 3 HA tags and Gateway sites were added to the original TRE_GOI_rtTA_hPGK vector [*Barde et al., 2006*], by mixing 100 ng of TF entry clone with 100 ng of the expression vector and 0.5 μl of LR clonase II enzyme mix (Life Technologies). After incubating 18 hr at 25°C, this mix was transformed into competent STBL3 cells. Successfully subcloned TFs (734) were miniprepped using the NucleoSpin 96 Plasmid miniprep kit (Macherey–Nagel), typically yielding a concentration of approximately 150–300 ng/μl, from 2 ml overnight cultures in terrific broth II (MP Biomedicals, Santa Ana, CA). We note that the adipogenic regulators C/EBPα and C/EBPβ were not successfully cloned.

## High-throughput lentiviral production (96 well plates)

Lentivirus production was done using the TF ORF Tet-On Gateway-compactible expression vectors. Briefly, 293T packaging cells were seeded at 0.2 millions of cells/ml (100 μl per well) in low-antibiotic growth medium (DMEM, 10% iFBS (HyClone), 0.1x Pen/Strep) in a 96-well tissue culture plate (Corning, Corning, NY). The cells were incubated until they reached 70% confluence. In a 96-well polypropylene storage plate (Corning), 100 ng TF ORF Tet-On expression vector, 100 ng psPAX2 and 10 ng pMD2.G plasmids were mixed with OPTI-MEM (Life Technologies) to a total volume of 25 μl per well. At the same time, lipofectamine 2000 Transfection Reagent (Life technologies) was mixed with OPTI-MEM in an eppendorf tube (0.5 μl lipofectamine with 24.5 μl OPTI-MEM per well) and incubated for 5 min at room temperature. The transfection reagent mix was subsequently transferred to each well of the 96-well plasmid plate and the plate was incubated for 20 min at room temperature. Before transfection, the medium in the 96-well culture plate was changed with a fresh low-antibiotic growth medium, 100 μl per well, and the transfection mix (50 μl per well) was carefully transferred to the packaging cells. The cells were incubated for 18 hr in a 5% $CO_2$ humidified atmosphere at 37°C. The medium was then replaced with 170 μl high-BSA growth medium (DMEM, 10% iFBS, 11 mg/ml microbiology-grade Bovine Serum Albumin (VWR Radnor, PA), 1x Pen/Strep). After 24 hr of incubation (37°C, 5% $CO_2$), 150 μl per well was collected and stored at −20°C.

## 3T3-L1 lentiviral differentiation assay in a 384-well plate

The differentiation assay was done in a collagen-coated 384-well black tissue culture plate for the overexpression screen. 3T3-L1 cells were seeded at a density of 20,000 cells per well (100 μl) and after

1 day, lentivirus was added to the medium (ratio 1:1 or 1:2 with the medium) with polybrene (8 µg/ml, Santa Cruz Biotechnology, Santa Cruz, CA). After 24 hr of incubation (37°C, 5% $CO_2$), the medium was replaced with complete DMEM (DMEM, 10% FBS, 1% Pen/Strep) to remove viruses. When confluence was reached, the medium was changed with complete DMEM, 1 µl/ml Doxycycline (hyclate from Sigma) and cells were incubated for 72 hr (37°C, 5% $CO_2$). Then, the cells were differentiated as described under 'Cell culture and differentiation', with addition of doxycycline in the medium until day 2. The cells were fixed and stained at day 7 of differentiation.

### Fixation, staining, and image analysis

The cells were washed with 100 µl 1x PBS per well and fixed by adding 1x PBS with 4% formaldehyde (Sigma). The cells were incubated at room temperature for 1 hr or overnight at 4°C. The fixation solution was then removed, cells washed once with 1x PBS and 100 µl of 1x PBS was added. Plates were then either stored at 4°C or stained immediately. Before staining, the cells were washed two times with water (100 µl). The staining of lipids with the lipophilic, fluorescent dye BODIPY, nuclei with Hoechst, and complete cells with SYTO60, the image acquisition (CellWorx, Applied Precision) and analyses were performed as described previously (*Meissburger et al., 2011*). The percentage of differentiation (PDC) for each TF was obtained by dividing the number of cells having at least four lipid droplets by the total number of cells. Fold changes (FCs) were calculated by comparison to the differentiation percentages obtained with the original Tet-on expression vector (47.9% (average) of differentiation, *Supplementary file 1A* [*Gubelmann et al., 2014*]). 307 TFs showing a FC > 1 are displayed in *Figure 1B* and those with a FC ≥ 1.5 (i.e., a 50% increase in the fraction of mature adipocytes) in three technical replicates at a FWER of ≤ 0.05 (each replicate normalized to negative controls using normal transform and normal two-sided p-values corrected for multiple testing with Bonferroni's correction) were considered significant enhancers of adipogenesis, are referred to as 'candidates' in the manuscript, and are highlighted in red or orange in *Figure 1B* and in green in *Supplementary file 1A* (*Gubelmann et al., 2014*).

### Analysis of candidates

We assessed which of the 26 candidates were endogenously transcribed in 3T3-L1 cells as well as human adipose stromal cells using publicly available data (*Nielsen et al., 2008*; *Mikkelsen et al., 2010*): mRNA levels (micro-array-based) as well as RNA POLII genome-wide binding (ChIP-seq). RNA POLII enrichment over gene bodies across all fat cell differentiation time points was used to determine whether genes were transcribed or not, based on a threshold adjusted by qPCR (10 sequence tags per 500 bp at the highest transcribed day; *data not shown*). We also used previously determined gene expression classes (*Nielsen et al., 2008*) based on mRNA levels during differentiation of both human and mouse pre-adipocytes. The three qualitative transcription measures are listed in *Supplementary file 1A* (*Gubelmann et al., 2014*) for each candidate and summarized in *Figure 1—figure supplement 1B*.

We used Expression Atlas (September 2013 version) to compare the expression of the candidates in adipogenesis (filters 'adipose' and 'adipose tissue') to their expression in other tissues. Nine and eight TFs, respectively, were significantly expressed higher and lower in adipose tissue compared to the average across all tissues (p = 0.05). This corresponds to a 1.5-fold enrichment of significantly higher expressed genes compared to random, an enrichment which is not significant (p = 0.1, 10,000 permutations).

### Lentiviral production and stable cell lines

Viral particles containing Tet-on or shRNA expression plasmids were generated in 293T cells as described previously (*Barde et al., 2010*), with slight modifications. Instead of five plates, one plate of 15 cm was transfected for each expression plasmid and the supernatant was harvested three times, every 8–12 hr. After the ultracentrifugation step (1 tube for each vector, 35 ml), the pellets were re-suspended in 35–45 µl of 1x PBS, centrifuged at maximum speed and stored at −80°C in 10 µl aliquots.

For stable and selected lines, 3T3-L1 cells at a density of $5 \times 10^4$ cells were transduced with 10 µl viral particles in a six-well plate. After 12 hr, the medium was changed and puromycin (Life Technologies, 2 µg/ml) was added after 72 hr to select stably transduced cells. If the cell confluence reached 80%, the cells were split and transferred into larger dishes. After every 2 days, puromycin selection media was changed and the stably transduced cells were selected for 2 weeks before performing actual experiments.

For the Tet-On Gateway-compatible expression vector, the original vector was used as a negative control. For the KD, the lentiviral mammalian vector pLKO.1 containing specific shRNAs (three shRNAs per target were used, listed below) along with the control shRNA (empty pLKO.1 plasmid) were obtained from Sigma. Finally, the shRNA 2 ZEB1 was not used in the shRNA pool of ZEB1 as the robustness of the cells after treatment was low.

ShRNA: shRNA 1 PPARγ (TRCN0000001657), shRNA 2 PPARγ (TRCN0000001658), shRNA 3 PPARγ (TRCN0000001660), shRNA 1 ZEB1 (TRCN0000235850), shRNA 2 ZEB1 (TRCN0000235851), shRNA 3 ZEB1 (TRCN0000235853).

As the KDs were not stable for ZEB1 in these generated cell lines, we directly infected 50,000 3T3-L1 cells/well in six-well plates (three replicates per constructs). The medium was changed the following day, and when the cells reached confluence, we proceeded with the adipocyte differentiation protocol.

## RNA isolation and quantitative PCR

Total RNA was isolated using a Qiagen RNAeasy plus mini kit according to the manufacturer's protocol by using either the RLT buffer lysis system or, for the RNA-seq samples, the TRIzol/Chlorophorm extraction procedure (Sigma, Saint-Louis, MO). The RNA concentration and quality was determined using a nanodrop ($1.8 \leq A260/A280 \leq 2.2$) and by visual inspection of separated bands on agarose gels. 2 µg of total RNA was used for single strand cDNA synthesis using the SuperScript VILO cDNA Synthesis Kit (Life Technologies). Then, cDNA was diluted 1:100 using nuclease free water and 1.5 µl was used for each qPCR reaction. Quantitative real-time PCR was performed in 384-well plates with three technical replicates on the ABI-7900HT Real-Time PCR System (Applied Biosystems) using the Power SYBR Green Master Mix (Applied Biosystems) using standard procedures. A Hamilton Liquid Handling Robotic System was used to assemble the 384-well plates. The qPCR primers were designed with in-house developed GETPrime software (*Gubelmann et al., 2011*) or taken from previous publications (*Supplementary file 1I* [*Gubelmann et al., 2014*]). They were checked for linearity and single product amplification.

## In vivo differentiation

The in vivo experiments were performed as described in *Meissburger et al. (2011)* by using the shRNAs listed above and the ZEB1 as well as PPARγ (positive control) overexpression clones. In short: fat tissue was dissected, minced and incubated in collagenase type II for 1 hr at 37°C. Approximately 106 cells were treated with virus and resuspended in 100 µl of Matrigel (BD Biosciences, San Jose, CA) before injection subcutaneously into a skin fold of the neck. After 6 weeks, Matrigel pads were excised. From each pad pictures of three full sections were taken and adipocyte numbers as well as the number of nuclei were determined automatically using the Cell Profiler Software. To overcome the doxycycline effect of a Tet-on inducible lentiviral vector, the TF ORFs were subcloned from the pDONR221 entry clones into the pLenti6/UbC/V5-DEST expression vector (Life Technologies, Carlsbad, CA). As negative controls, the original vector and the shEMPTY (Sigma) were used. All experiments were performed in three replicates and the significance of the observed changes was estimated using a one-sided Wilcoxon rank-sum test.

## Ex vivo differentiation of human SVF

Differentiation of the pre-adipocytes was induced as described previously (*Meissburger et al., 2011*). RNA were extracted after day 8 of differentiation and analyzed by qPCR, as explained above. mRNA expression was normalized to *36B4*. For expression of *Zeb1*, TaqMan gene expression assays (Ambion Life Technologies, Carlsbad, CA) were used according to the manufacturer's protocol. Differentiated cells were co-stained with adiponectin to verify differentiation.

## RNA-seq

After isolation of RNA, as described in the section 'RNA isolation and quantitative PCR', the Illumina Truseq RNA Sample Preparation kit v2 protocol (Illumina, San Diego, CA) was followed using 500 ng of RNA per sample as starting material. Half of the ligated reaction volume was used for PCR (14 cycles) and the other half was kept at −80°C as a backup. Libraries were checked for quality and quantified using the Bioanalyzer 2100 (Agilent, Santa Clara, CA), before being sequenced in barcoded pools of 16 samples on the Illumina Hiseq 2500 instrument (100 base paired-end sequencing, 4 lanes; Genomics Sequencing Facility, Nestle, Lausanne).

## RNA-seq analysis

Sequenced tags were aligned to Ensembl 70 gene annotation of the NCBI38/mm10 genome using Bowtie 1.0.0 and the parameters '-a --best --strata -S -m 100 --chunkmbs 256 -p 8'. Expression levels per transcript and gene were estimated using mmseq-1.0.2 (*Turro et al., 2011*) with default parameters. Quantile normalized expression estimates were transformed into pseudo-counts by un-logging, un-standardising and multiplying with gene length. Expression differences between the samples were quantified with DESeq (*Anders and Huber, 2010*), FC ≥ 1.5 and *padj* ≤ 0.01. Estimated FCs were validated by using qPCR-measured FCs at days 0 and 2, revealing significant correlations with Pearson's r > 0.95 and p = $2 \times 10^{-4}$ and p = $3 \times 10^{-5}$, respectively.

## Chromatin Immunoprecipitation

3T3-L1 stably selected cells were collected at days −2, 0, 2 and 4 for ZEB1. The cells were fixed as described previously (*Raghav et al., 2012*) and stored at −80°C. Ten million cells were used for each immunoprecipitation (IP). The ChIP experiment was performed as described previously in *Raghav et al. (2012)*, under 'Chromatin Immunoprecipitation of SMRT'. A ZEB1 antibody (Santa Cruz, sc-25388, 10 µg per IP) and a rabbit isotype control IgG (Santa Cruz Biotechnology, Santa Cruz, CA, sc-8994, 10 µg per IP) was used for each time point. The DNA was stored at −20°C until verification of ChIP enrichment by qPCR and ChIP-seq library preparation.

For the ChIP of ZEB1-HA and C/EBPβ, the chromatin samples were incubated overnight at 4°C with an anti-HA antibody (Abcam, UK, ab9110, 5 µg per IP), an anti-C/EBPβ antibody (Santa Cruz, sc-150, 10 µg per IP) respectively, or a rabbit control IgG (Santa Cruz, SC-8994) coupled to magnetic beads (sheep anti-rabbit IgG, Life Technologies, Carlsbad, CA Dynabeads), as described previously (*Kilpinen et al., 2013*) with few modifications. 50 µl of antibody-coupled beads were added to each 1 ml chromatin material instead of 100 µl. After incubation, we washed the beads five times with a LiCl wash buffer (100 mM Tris at pH 7.5, 500 mM LiCl, 1% NP-40, 1% sodium deoxycholate), mixed for 10 min at 4°C and removed remaining ions with a single wash with 1 ml of TE (10 mM Tris–HCl at pH 7.5, 0.1 mM Na2EDTA) at 4°C for 1 min mix. The beads were then resuspended in 200 µl IP Elution Buffer (1% SDS/0.1 M NaHCO3), incubated in a 65°C shaker for 1 hr and placed on the magnet to recover the supernatant. The supernatant was incubated at 65°C overnight to complete the reversal of the formaldehyde cross-links. The next day, DNA was purified from the reverse-crosslinked chromatin by proteinase and RNase digestion followed by purification using Qiagen DNA purification columns.

## ChIP-seq

Multiplex libraries were prepared using barcoded adapters for each sample following the protocol described in *Raghav and Deplancke (2012)* with slight modifications. In brief, ChIP-DNA fragments were end-repaired using an End-IT DNA end repair kit (Epicentre Technologies) followed by the addition of an A-base and ligation of bar-coded adapters. After the ligation incubation, the DNA was cleaned up using Agencourt AMPure XP Beads (Beckman Coulter, Fullerton, CA) and eluted in 12 µl. These purified, ligated DNA fragments were separated on a 2% agarose gel to select 200–500 bp-sized DNA fragments and DNA was subsequently isolated from gel slices using a Qiagen gel extraction kit. The gel-extracted DNA was then amplified for 17 cycles by PCR using high fidelity Phusion hot start polymerase (NEB, Ipswich, CA). The concentration and quality of purified, amplified DNA were estimated using respectively a Qubit dsDNA high sensitivity kit (Life Technologies) and a high sensitivity DNA assay Bioanalyzer 2100 (Agilent). After quality confirmation, the DNA libraries were sequenced on an Illumina High Seq 2500 (Genomics Sequencing Facility, Nestle, Lausanne; Genomic Technologies Facility, UNIL, Lausanne). Pass-filtered reads from the Illumina analysis pipeline were used for further analysis.

## Absolute quantification of ZEB1

### Peptide selection

For ZEB1 quantification, the following three proteotypic peptides were selected: VAVDGNVIR, AYYALNAQPSTEELSK and IADSVNLPLDGVK. Nuclear extracts (NE) at differentiation time points: day 0, 2 hr (2 hrs), day 2 and day 4 were analyzed as three biological and three technical replicates (*n* = 3). A total amount of 200 µg of NE was used for SDS-PAGE analysis followed by in gel trypsin digestion of the target protein. Heavy labelled (lysine, K) and accurately quantified synthetic peptide standards (JPT SpikeTides, JPT Peptide Technologies GmbH, Germany) were added to the digested NE at a final

concentration of 5 fmol/µl. Relative levels of RXRα protein were monitored by SRM along with ZEB1 protein as a quality control for all analyzed NE. The following three proteotypic peptides for RXRα: GLSNPAEVEALR, ILEAELAVEPK, and VLTELVSK, were selected and monitored by SRM as described previously (*Simicevic et al., 2013*).

## Liquid chromatography

Dried NE peptides were resuspended in 30 µl LC-MS/MS loading solvent (5% acetonitrile, 0.1% formic acid). Following resuspension, samples were allowed to settle for 1 hr to increase overall peptide solubility before SRM analysis. Typically, 5 µl of sample was loaded and captured on a homemade capillary pre-column (C18; 3 µm, 200 Å; 2 cm × 250 µm) before analytical LC separation (nanoACQUITY UPLC, Waters). Samples were separated using a 80-min biphasic gradient starting from 100% solvent A (2% acetonitrile, 0.1% formic acid) to 90% solvent B (100% acetonitrile, 0.1% formic acid) on a Nikkyo (Nikkyo Technology, Japan) nanocolumn (C18; 3 µm, 100 Å; 15-cm length and 100-µm inner diameter; flow of 0.5 µl/min). The gradient was followed by a wash for 8 min at 90% solvent B and column re-equilibration for 12 min at 100% solvent A.

## Selected reaction monitoring (SRM) of ZEB1 peptides

All samples were analyzed on a TSQ-Vantage triple quadrupole mass spectrometer (Thermo Fisher Scientific, Waltham, MA). A 0.7-FWHM-resolution window for both Q1 and Q3 was set for parent- and product-ion isolation. Fragmentation of parent ions were performed in Q2 at 1.5 mtorr, using collision energies calculated with the Pinpoint software (v.1.3). Parent-ion selection was set for fully digested peptides of the doubly charged ion for target peptides. Peptide fragment transitions and collision energies were established using the heavy labelled synthetic reference peptides (JPT SpikeTides). Generally, singly charged peptide fragment ions ranging from $y_3$ to $y_{n-1}$ were monitored. A parent-ion retention-time target window of 2 min was selected for all proteins monitored during a scheduled SRM ru*n*. A total of 79 transitions were monitored during a single LC-SRM run, using a cycle time of 0.5 s and a minimum dwell time of 10 ms. All data analyses were carried out using Pinpoint software (v.1.3). Peptide identification and peak-area integration of all targeted peptides as well as their transitions were manually verified in Pinpoint.

While absolute ZEB1 levels fluctuated about two-fold between biological replicates likely due to technical variations in the levels of the targeted peptides, the overall trend was consistent between biological replicates (*Figure 2A* and *Figure 2—figure supplement 1D*).

## ZEB1 immunoprecipitation and mass spectrometry

For IP of ZEB1 protein complexes, 3T3-L1 cells stably overexpressing ZEB1 and control vectors were grown in 150-mm plates and ZEB1 expression was induced at day −2 (the time of cell confluence) using doxycycline as described before. At day 0 of differentiation, the cells were washed with 1x PBS and dissociated using trypsin-EDTA solution (Life Technologies). The dissociated cells were collected in 15-ml falcon and centrifuged at 250 × g to pellet the cells. The cell pellet was washed once with 1x PBS containing 1 mm PMSF (phenylmethylsulfonyl fluoride) protease inhibitor and stored at −80°C. Before lysing the cells for IP, recombinant protein-A sepharose beads (Life Technologies) were prepared by washing with IP buffer (20 mM Tris-Cl, pH. 7.4, 150 mM NaCl, 10% glycerol, 1% Triton X-100, 1 mM EDTA, 1 mM DTT) supplemented with protease and phosphatase inhibitors (Roche). The anti-rabbit HA tag antibody (Abcam, UK) was conjugated with the protein-A sepharose beads by incubating antibody with the beads overnight at 4°C at rotation. The cells were then lysed in 1 ml IP buffer by douncing 20-25 times using 1 ml syringe. Lysates were cleared by centrifuging at maximum speed in a tabletop refrigerated centrifuge for 10 min. The tagged complexes were pulled down using 50 µl HA-antibody tagged beads for 4 hr at 4°C at rotation. The beads were then washed 5 times using IP buffer and the bound protein complexes were eluted by heating the beads in 60 µl 2x Laemmli sample buffer. To resolve the proteins present in pull down complexes, a 12% SDS-PAGE gel was used.

Entire lanes of SDS-PAGE gels were then sliced into pieces. Samples were first washed twice for 20 min in 50% ethanol and 50 mM ammonium bicarbonate (AB). Gel slices were dried down by vacuum centrifugation. All samples were reduced/alkylated using dithioerythritol and iodoacetamide. Gel pieces were dried again and re-hydrated using trypsin solution (12.5 ng/µl in 50 mM AB and 10 mM CaCl2). Trypsin digestion was performed overnight and resulting peptides were extracted twice for 20 min in 70% ethanol and 5% Formic Acid (FA). Samples were dried down and re-suspended in 2%

acetonitrile and 0.1% FA for LC-MS/MS injections. One-dimensional liquid chromatography separation was performed on a Dionex Ultimate 3000 RSLC nano UPLC system (Dionex, Sunnyvale, CA) on-line connected with an Orbitrap Q Exactive Mass-Spectrometer (Thermo Fischer Scientific). A self-made capillary pre-column (Magic AQ C18; 3 µm-200 Å; 2 cm × 100 µm ID) was used for sample trapping and cleaning. Analytical separation was then performed using a C18 capillary column (Nikkyo Technos Co, Japan; Magic AQ C18; 3µm-100 Å; 15 cm × 75 µm ID) at 250 nl/min. Separation of peptides was carried over an 85 min biphasic gradient. Mass spectrometric measurements were performed using a data-dependent top 20 method, with the full-MS scans acquired at 70 K resolution (at m/z 200) and MS/MS scans acquired at 17.5K resolution (at m/z 200). A database search was performed using Mascot 2.3 (Matrix Science) and SEQUEST in Proteome Discoverer v.1.3 against a mouse database (UniProt release 2013_09). All searches were performed with trypsin cleavage specificity, up to two missed cleavages were allowed and ion mass tolerance of 10 ppm for the precursor and 0.05 Da for the fragments. Carbamidomethylation was set as a fixed modification, whereas oxidation (M), acetylation (Protein N-term), phosphorylation (STY) were considered as variable modifications. Data were further processed and inspected in the Scaffold 4 software (Proteome Software, Portland, OR).

## ChIP-seq analysis

Sequenced ChIP tags were aligned to the NCBI38/mm10 genome with Bowtie2 (*Langmead and Salzberg, 2012*) and the parameters '--very-sensitive -M 10 -p 8'. Duplicates were removed, as well as the reads with a mapping quality under 10. Regions showing significant ChIP enrichment (peaks) with CCAT 3.0 (*Xu et al., 2010*) ('fragmentSize 200 slidingWinSize 100 movingStep 50 isStrandSensitiveMode 1 minCount 13 minScore 5 bootstrapPass 80', FDR 0.1) or Homer (*Heinz et al., 2010*) ('-F 3 -L 4 -localSize 5000 -C 4 -fragLength 200 -minDist 500 -center') using anti-HA (in Zfp277-HA overexpressing cells, where Zfp277-HA serves as negative control as it is not imported into the nucleus, *data not shown*) as control were merged into common ZEB1-bound regions. This resulted in a total of 43,405 ZEB1 bound regions (10,708 day −2, 27,854 day 0, 18,145 day 2, 22,249 day 4) and 19,055 ZEB1-HA bound regions (day 0). Read counts contained in the genomic intervals defined by day 0 and merged ZEB1 binding, respectively, were correlated and the Spearman's ρ used to generate hierarchically clustered heatmaps (*Figure 3—figure supplement 1B–C* and *Figure 4—figure supplement 1B*). As biological replicates were only available for days −2 and 0, we compared the read counts contained in the genomic intervals defined by merged ZEB1 binding at early time points (two replicates day −2 and high enrichment replicate day 0 treated as replicates) to those at late time points (single replicate day 2 and single replicate day 4 treated as replicates) using DESeq 1.2.3. We obtained 803 regions showing significantly (*padj* ≤ 0.1, FC ≥ 2) higher ZEB1 ChIP-seq read counts at days 2 and 4 vs. days −2 and 0 (*late-only* ZEB1 binding) and 552 regions showing significantly higher ZEB1 ChIP-seq read counts at days −2 and 0 vs. days 2 and 4 (*early-only* ZEB1 binding). Genes overlapping at least one late-only bound region (gene body +500 bp upstream of TSS) and no early-only bound region were considered 'late-only genes'; conversely, genes overlapping at least one early-only bound region (gene body +500 bp upstream of TSS) were considered 'early-only genes'. Genes overlapping only statically ZEB1-bound regions were considered 'static-only genes'.

## Motif discovery

Motif discovery was performed with HOMER's findMotifsGenome (*Heinz et al., 2010*) ("-size 100 -len 6,8,10,12,14 -local") on all ZEB1-bound regions at day 0 (summary results are displayed in *Figure 3D* and complete results listed in *Supplementary file 1E* [*Gubelmann et al., 2014*]) and with MEME 4.9.1 (*Bailey et al., 2006*) ('-nmotifs 10 -minsites 10 -minw 4 -maxw 20 -revcomp -maxsize 60,000 -dna') using 50 bp centered on the summits of the most highest-scoring 1000 ZEB1-bound regions at day 0 (the top enriched motif is displayed in *Figure 3C*). Motif matching to known motif databases was performed using TOMTOM 4.9.1 (*Bailey et al., 2006*) and 'All vertebrates' database. Motif scanning of bound regions (100 bp centered around the summit of ZEB1-bound regions at day 0 as well as randomly shifted regions for background comparison) was performed with the PWMs available through *Wang et al. (2012)* and Jaspar (*Bryne et al., 2008*) for ZEB1, CEBPB, AP1, NFIC and SMAD3 using the package Biostrings 2.30.0 at a cutoff of 85%. Percentage of peaks showing at least one and at least two PWM matches are displayed in *Figure 3—figure supplement 1F*. ZEB1 PWM matches in an 800 bp region centered on ZEB1 peak summits (at day 0) were displayed as a motif density plot in *Figure 3—figure supplement 1E*. Differential motif discovery contrasted early-only and late-only

ZEB1-bound regions with static ($padj$ > 0.1, FC < 2) ones. Early-only ZEB1 binding enriched for RUNX (MEME E = 7 × 10$^{-29}$, matching RUNX2/MA0511.1, TOMTOM p = 2 × 10$^{-6}$ and RUNX1/MA0002.2, TOMTOM p = 7 × 10$^{-6}$) and TEAD1 (MEME E = 2 × 10$^{-7}$, matching TEAD1/MA0090.1, TOMTOM p = 5 × 10$^{-6}$) motifs, while late-only binding enriched for C/EBP (MEME E = 9 × 10$^{-179}$, matching C/EBPα / MA0102.3, TOMTOM p = 1 × 10$^{-12}$ and C/EBPβ/MA0466.1, TOMTOM p = 2 × 10$^{-7}$), NFI (MEME E = 4 × 10$^{-12}$, matching NFIC/MA0161.1, TOMTOM p = 9 × 10$^{-5}$ and TLX::NFIC/MA0119.1, TOMTOM p = 1 × 10$^{-3}$) and PPARG::RXR (MEME E = 3 × 10$^{-6}$, matching PPARG::RXR/MA0065.2, TOMTOM p = 4 × 10$^{-7}$) motifs, as displayed in *Figure 4C* and *Figure 4—figure supplement 1D*.

## Annotation and Gene Ontology enrichment

ZEB1 bound regions and regulated genes were annotated using Ensembl 70. Peaks were assigned (in this order) to either TSS (± 500 bp of annotated TSS), exonic regions (but not TSS), intronic regions (but not TSS or exons), gene proximity (10 kb distance to a gene, but not TSS or gene), or gene-distal regions (none of the above) and the fraction belonging to these categories displayed as pie charts in *Figure 3B*. For comparison, we performed the same analysis for randomly shifted ZEB1, C/EBPβ (day 0), and POLII (day 0) bound regions (*Figure 3—figure supplement 1D*). Additionally, we used HOMER's annotatePeaks with default options to assign each peak to its closest gene and report the p-values obtained using the hypergeometric test in the manuscript as indicative of a significant association of ZEB1 binding with promoters, CpG islands and exons. Complete results are included in *Supplementary file 1D* (*Gubelmann et al., 2014*). GO term enrichment analysis of ZEB1-regulated genes was performed with GeneGO MetaCore (https://portal.genego.com/), representative summaries are displayed in *Figure 2C* and complete results included in *Supplementary file 1C*. GO term enrichment analysis on ZEB1-bound regions was performed with GREAT using default parameters (*McLean et al., 2010*), contrasting late-only and early-only regions with static ones. Selected results are displayed in *Figure 4D* and *Figure 4—figure supplement 1E* and complete ones are included in *Supplementary file 1G* (*Gubelmann et al., 2014*).

## Publicly available data

We used the following publicly available data: (1) in 3T3-L1 cells: microarray-based gene expression, ChIP-seq using antibodies against JUNB, FOSL, CJUN, ATF7, ATF2, C/EBPβ, PPARγ, POLII, H3K9AC, RXRα, DNAse-seq (*Nielsen et al., 2008*; *Steger et al., 2010*; *Siersbæk et al., 2011*; *Raghav et al., 2012*; *Siersbæk et al., 2014*); (2) pre-adipocyte vs. adipocyte data and meta-analysis of mouse tissue expression data available through Array Express (September 2013) (*Rustici et al., 2013*); (3) Human ENCODE HepG2 and lymphoblastoid cell lines (LCLs): ChIP-seq with antibodies against C/EBPβ, FOSL, JUN and ZEB1 (*ENCODE Project Consortium, 2012*). All ChIP-seq and DNase-seq data were reanalyzed analogous to the in-house generated data. The human data was aligned to the GRCh37 (hg19) genome. Overlaps between regions bound by ZEB1 (or randomly shifted ZEB1) and C/EBPβ at day 0 as well as AP1 factors (JUNB, FOSL, CJUN, ATF7 and ATF2) at 4 hr (day 0) are displayed as Venn diagrams using the R package VennDiagrams 1.6.5 in *Figure 3E* and *Figure 3—figure supplement 1G*.

## Heatmaps and genomic loci displays

Read counts were divided by normalization factors estimated using DESeq2 and shifted ZEB1 peaks as genomic intervals and extended to 200 bp each. Counts were then summed across 400 windows of 5 bp each (total 2 kb) centered around ZEB1 peak summits and log2 transformed. These transformed values (referred to as 'norm ChIP' in *Figures 3F and 4B*, *Figure 3—figure supplement 1H*, and *Figure 4—figure supplement 1A*) were displayed as heatmaps using the R package pheatmap 0.7.7. We note that only a subset (top, mid and bottom 4000 peaks sorted by mean ZEB1 day 0 ChIP enrichment) is displayed in *Figure 3—figure supplement 1H* and only the two AP1 factors showing the highest colocalization (data not shown) with ZEB1—ATF2 and ATF7—are displayed in *Figure 3F* and *Figure 3—figure supplement 1H*. Similarly, counts for the human ChIP-seq datasets were normalized to the library size, extended to 200 bp and log2 transformed. Mean values in a 8 kb region centered on ZEB1 peak summits in lymphoblastoid cell lines (LCLs) were then plotted in *Figure 3—figure supplement 1I*. Genomic loci plots were visualized in the UCSC Genome Browser based on bedGraph files obtained using HOMER's makeUCSCfile and the parameters '-o auto -res 1 -fsize 5e7'. The scale used for each individual track is displayed in *Figures 3A and 4A*, *Figure 3—figure supplement 2E* and *Figure 4—figure supplement 1C*.

## Adipogenic gene regulatory network

We used Wikipathways, as well as recent reviews to manually compile an adipogenic transcriptional regulatory network (*Kelder et al., 2012*; *Rosen and MacDougald, 2006*; *Siersbæk et al., 2012*) and displayed it using Pathvisio (*van Iersel et al., 2008*). We then superimposed the expression information (log2 FCs and multiple-testing corrected p-values after ZEB1 KD at day 0) as well as ZEB1 and C/EBPβ binding information (at day 0: overlapping TSS or overlapping the gene) on the network in *Figure 5A*.

## Bioethics

All mouse experiments were conducted in strict accordance with Swiss law and all experiments were approved by the ethics commission of the state veterinary office (60/2012, 43/2011). The work on obese subjects was approved by the ethics committee at the University Hospital of Heidelberg and is conforming to the ethical guidelines of the 2000 Helsinki declaration. All participants provided witnessed written informed consent prior to entering the study (S-365/2007). The trial was registered as NCT00773565.

## Data access

ChIP-seq data are available in the ArrayExpress database (www.ebi.ac.uk/arrayexpress) under accession number E-MTAB-2537. RNA-seq data are available in the ArrayExpress database under accession number E-MTAB-2538. Results of the TF overexpression screen, processed RNA-seq data, mass spectrometry results and the clinical data (*Supplementary file 1A*, *Supplementary file 1B*, *Supplementary file 1F* and *Supplementary file 1H*) have additionally been deposited in the Dryad data repository under doi: 10.5061/dryad.j966f (see *Gubelmann et al., 2014*).

## Acknowledgements

We thank Paola Gilardoni, Adrian Schmid and Romain Hamelin for their support in performing the mass spectrometry experiments, the members of the Genomic Technologies Facility (UNIL) and VITAL-IT for respectively performing the Illumina sequencing and accommodating biocomputation; Johan Auwerx, Carlos Alvarez, and Gerhard Christofori for helpful discussions and proofreading and members of the Deplancke lab for technical support.

## Additional information

### Funding

| Funder | Grant reference number | Author |
|---|---|---|
| Swiss National Science Foundation | 31003A_122552, 31003A_138323 | Carine Gubelmann, Sebastian M Waszak, Bart Deplancke |
| Federation of European Biochemical Societies | | Petra C Schwalie |
| European Molecular Biology Organization | ALTF 600-2013 | Petra C Schwalie |
| Human Frontier Science Program | LT001032/2013-L | Petra C Schwalie |
| École Polytechnique Fédérale de Lausanne | | Bart Deplancke |
| European Research Council | 205641 AdipoDif | Christian Wolfrum |
| Swiss National Science Foundation | 31003A_140926 | Christian Wolfrum |

The funders had no role in study design, data collection and interpretation, or the decision to submit the work for publication.

### Author contributions

CG, PCS, SKR, Conception and design, Acquisition of data, Analysis and interpretation of data, Drafting or revising the article; ER, TD, GU, Conception and design, Acquisition of data, Analysis and

interpretation of data; EK, Acquisition of data, Analysis and interpretation of data; SMW, Analysis and interpretation of data, Drafting or revising the article; AC, GR, Conception and design, Acquisition of data, Analysis and interpretation of data, Contributed unpublished essential data or reagents; WH, Conception and design, Acquisition of data; DT, Drafting or revising the article, Contributed unpublished essential data or reagents; CW, BD, Conception and design, Analysis and interpretation of data, Drafting or revising the article

## Author ORCIDs

Sebastian M Waszak, http://orcid.org/0000-0003-3042-9521

## Ethics

Human subjects: The work on obese subjects was approved by the ethics committee at the University Hospital of Heidelberg and is conforming to the ethical guidelines of the 2000 Helsinki declaration. All participants provided witnessed written informed consent prior entering the study (S-365/2007). The trial was registered as NCT00773565. Animal experimentation: All animal experiments were conducted in strict accordance with Swiss law and all experiments were approved by the ethics commission of the state veterinary office (60/2012, 43/2011).

# Additional files

## Supplementary files

• Supplementary file 1. *Gubelmann et al. (2014)*. (**A**) Transcription factor screen. Percentage differentiated cells in response to TF overexpression. Raw values, fold-changes with respect to control, p-values and associated annotations are included for all TFs showing positive effects on adipogenesis. TFs significantly enhancing differentiation are highlighted in green (FC ≥ 1.5, Bonferroni $\alpha = 0.05$). (**B**) mRNA levels upon ZEB1 knockdown. Estimated expression levels in all RNA-seq replicates (shZEB1 and shEmpty at days 0 and 2), fold-changes and p-values as well as information ZEB1 and C/EBPβ binding at gene TSS, gene bodies or in gene proximity. (**C**) Pathway Enrichments. Pathways enriched in genes significantly de-regulated upon ZEB1 knockdown at days 0 and 2. (**D**) Gene Ontology Enrichments Genomic features enriched at ZEB1-bound locations. (**E**) Motifs in ZEB1 bound regions. Motifs enriched in 100 bp centered on ZEB1 peak summits. (**F**) Mass Spectrometry results. List of proteins detected by at least one peptide in any of the two performed ZEB1 immunoprecipitation experiments. (**G**) GREAT Gene Ontology enrichments. GO Terms enriched in genes proximal to late-only and early-only ZEB1-bound regions. (**H**) Clinical data: Correlation of *Zeb1* and *Rorg* mRNA levels in human obese subjects with a range of adipogenesis-relevant measures. Raw values are included for the significant correlations discussed in the manuscript. (**I**) qPCR primers. List of primers used for gene expression analyses and ChIP-qPCR.

## Major datasets

The following datasets were generated:

| Author(s) | Year | Dataset title | Dataset ID and/or URL | Database, license, and accessibility information |
| --- | --- | --- | --- | --- |
| Gubelmann Carine, Schwalie Petra C, Raghav Sunil K, Röder Eva, Delessa Tenagne, Kiehlmann Elke, Waszak Sebastian M, Corsinotti Andrea, Udin Gilles, Holcombe Wiebke, Rudofsky Gottfried, Trono Didier, Wolfrum Christian, Deplancke Bart | 2014 | ZEB1 in 3T3-L1 differentiation (ChIP-seq), E-MTAB-2537 | https://www.ebi. ac.uk/arrayexpress | Publicly available at EBI ArrayExpress. |
| Gubelmann Carine, Schwalie Petra C, Raghav Sunil K, Röder Eva, Delessa Tenagne, Kiehlmann Elke, Waszak Sebastian M, Corsinotti Andrea, Udin Gilles, Holcombe Wiebke, Rudofsky Gottfried, Trono Didier, Wolfrum Christian, Deplancke Bart | 2014 | ZEB1 in 3T3-L1 differentiation (RNA-seq), E-MTAB-2538 | https://www.ebi. ac.uk/arrayexpress | Publicly available at EBI ArrayExpress. |

| Gubelmann Carine, Schwalie Petra C, Raghav Sunil K, Röder Eva, Delessa Tenagne, Kiehlmann Elke, Waszak Sebastian M, Corsinotti Andrea, Udin Gilles, Holcombe Wiebke, Rudofsky Gottfried, Trono Didier, Wolfrum Christian, Deplancke Bart | 2014 | Data from: Identification of ZEB1 as a central component of the adipogenic gene regulatory network | 10.5061/dryad.j966f | Available at Dryad Digital Repository under a CC0 Public Domain Dedication. |

The following previously published datasets were used:

| Author(s) | Year | Dataset title | Dataset ID and/or URL | Database, license, and accessibility information |
|---|---|---|---|---|
| Stunnenberg HG, Hagenbeek D, Moulos P, Franöoijs K, Denissov S, Megens E, Mandrup S, Nielsen C, öskov Pedersen T, Siersbæk R, Børgesen M | 2008 | Genome-wide profiling of PPARγ:RXR and RNA polymerase II | GSE13511; http://www.ncbi.nlm.nih.gov/geo/ | Publicly available at NCBI Gene Expression Omnibus. |
| Siersbaek R, Nielsen R, John S, Sung M, Baek S, Loft A, Hager GL, Mandrup S | 2011 | Extensive Chromatin Remodeling and Establishment of Transcription Factor andapos;Hotspotsandapos; during Early Adipogenesis | GSE27826; http://www.ncbi.nlm.nih.gov/geo/ | Publicly available at NCBI Gene Expression Omnibus. |
| Myers R, Pauli F | 2011 | Transcription Factor Binding Sites by ChIP-seq from ENCODE/HAIB | GSE32465; http://www.ncbi.nlm.nih.gov/geo/ | Publicly available at NCBI Gene Expression Omnibus. |
| Mikkelsen TS, Xu Z, Gimble JM, Lander ES, Rosen ED | 2010 | Expression profiling of 3T3-L1 adipogenesis | GSE20696; http://www.ncbi.nlm.nih.gov/geo/ | Publicly available at NCBI Gene Expression Omnibus. |
| Steger DJ, Grant GR, Schupp M, Tomaru T, Lefterova MI, Schug J, Manduchi E, Stoeckert CJ Jnr, Lazar MA | 2010 | Propagation of Adipogenic Signals through an Epigenomic Transition State | GSE21898; http://www.ncbi.nlm.nih.gov/geo/ | Publicly available at NCBI Gene Expression Omnibus. |
| Siersbæk R, Rabiee A, Nielsen R, Sidoli S, Traynor S, Loft A, Poulsen LL, Rogowska-Wrzesinska A, Jensen ON, Mandrup S | 2014 | Transcription factor cooperativity in early adipogenic hotspots and super-enhancers | GSE56872; http://www.ncbi.nlm.nih.gov/geo | Publicly available at NCBI Gene Expression Omnibus. |

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
