## [Decision Letter]

Thank you for sending your work entitled “Identification of ZEB1 as a central component of the adipogenic gene regulatory network” for consideration at *eLife*. Your article has been favorably evaluated by Fiona Watt (Senior editor) and 3 reviewers, one of whom is a member of our Board of Reviewing Editors.

The Reviewing editor and the other reviewers discussed their comments before we reached this decision, and the Reviewing editor has assembled the following comments to help you prepare a revised submission.

The reviewers were in agreement that the work was exciting and potentially of interest to the readers of *eLife*. However, several major issues were identified that need to be addressed in order to strengthen the conclusions drawn. These issues are summarized below:

1) Clarification regarding the screening system. Under standard conditions, close to all 3T3-L1 cells typically differentiate. That suggests that the setup used for the transcription factor screen somehow affects the differentiation capacity of the control cells, which allows one to identify factors that result in an increase in the differentiation capacity. This is supported by the relatively low degree of differentiation of the control cells in Figure 1 compared to control cells in Figure 1. This should be clarified in the manuscript. Microscopic pictures of the differentiation capacity of control cells would be valuable for evaluation of the degree of differentiation. Also, based on the Western blotting in S1c, it isn't clear what is being overexpressed. Why are there so many HA-tagged products of such varying sizes?

2) Effect of ZEB1 knockdown on ZEB1 promoter occupancy. The ChIP-seq for Zeb1 should have a knockdown control. There are a large number of weak peaks, most of which seem not to have the ZEB1 motif. How many of these peaks are still present in the absence of ZEB1?

3) Relationship of ZEB1 and C/EBPb. The authors convincingly show that ZEB1 binds to many chromatin regions together with C/EBPβ and this factor also associates with ZEB1 in a co-IP experiment indicating that these factors cooperate on chromatin. This conclusion would be strengthened if it could be demonstrated that C/EBPβ binding to chromatin is affected by ZEB1 knockdown and vice versa.

4) Better correlation between gene expression and promoter occupancy. The correlation between ChIP-seq data and gene expression profiles is well documented (Figure 4 and supplemental information). However, it would be informative if the RNA-seq data upon ZEB1 knock down were included here to make it clear whether genes associated with early, late, or static ZEB1 binding are in fact affected by ZEB1 knock down. The data is already used in Figure 2, so it should be straightforward to analyze the effect of knock down on the expression of putative direct ZEB1 target genes identified by ChIP-seq.

5) Additional data for the in vivo studies. For the in vivo differentiation of implanted precursor cells, it would be relevant to know whether ZEB1 knockdown affects proliferation of precursor cells, their commitment or just their maturation to adipocytes. Also, data on gene expression would be informative.

6) Clarification of the human data. The human data in Figure 7 is somewhat at odds with prior results. Is adiponectin positively correlated with body weight in these samples? It looks like it might be, which is not generally how those two variables interact.

7) Claims regarding commitment are premature. A weak point in the manuscript is the data on the importance of ZEB1 for commitment of MSCs to the adipocyte lineage. Although the authors provide evidence that ZEB1 is required for terminal differentiation of MSCs (i.e. C3H10T1/2 cells), it is not clear if this is because ZEB1 is required for the commitment step. The analysis of expression of Zfp423 and Zfp521 upon ZEB1 knock down is not convincing in itself. If the authors want to claim that ZEB1 also regulates commitment, this should be more thoroughly evaluated experimentally. Note, the editor believes that this question is better suited for a future manuscript and strongly suggests that the authors simply soften their conclusions regarding a role in commitment in the present paper. Alternatively, if the authors wish to make this claim, additional studies would be required. Some of the questions that could be addressed include: (1) can over expression of ZEB1 substitute for BMP-2/4 signaling to commit C3H10T1/2 cells to the adipocyte lineage?; (2) does overexpression of ZEB1 decrease the potential of C3H10T1/2 cells to develop into other cell lineages?; (3) is transient overexpression or knock down during commitment enough to enhance or decrease, respectively, differentiation of C3H10T1/2 cells?; or (4) is overexpression or knock down only during terminal differentiation of C3H10T1/2 enough to enhance or inhibit differentiation, respectively?

---

## [Author Response]

*1) Clarification regarding the screening system. Under standard conditions, close to all 3T3-L1 cells typically differentiate. That suggests that the setup used for the transcription factor screen somehow affects the differentiation capacity of the control cells, which allows one to identify factors that result in an increase in the differentiation capacity. This is supported by the relatively low degree of differentiation of the control cells in*
Figure 1
*compared to control cells in*
Figure 1*. This should be clarified in the manuscript. Microscopic pictures of the differentiation capacity of control cells would be valuable for evaluation of the degree of differentiation. Also, based on the Western blotting in S1c, it isn't clear what is being overexpressed. Why are there so many HA-tagged products of such varying sizes?*

In our study we used a differentiation protocol without rosiglitazone, which does not result in 100% differentiation of 3T3-L1 cells. We implemented this protocol because we were specifically interested in identifying transcription factors that enhance adipogenesis. If rosiglitazone had been used, then the margin to see this effect would have been very low since PPAR activators such as rosiglitazone yield close to complete differentiation of 3T3-L1 cells (e.g. Chawla and Lazar, PNAS, 1994). With our protocol, the average differentiation of cells was close to 50%, leaving ample room for observing enhancing effects. To be completely clear, but considering the fact that the method has already been described several times in the literature (e.g. Meissburger et al. EMBO Mol Med, 2011; Trajovski & Stoffel, Nat Cell Biol, 2012), we included microscopic images of our differentiation screen in Figure 7. If it is deemed necessary, these images can also be included in the supplementary material of the manuscript*.*Author response image 1.(**A**) Microscopic images of BODIPY (lipids, in green) and Hoechst (nuclei, in red)-stained 3T3-L1 cells transduced with control, ZEB1 and PPARG overexpression constructs. (**B**) Adiponectin levels plotted against fat mass, weight and BMI of obese subjects. Spearman’s ρ and p-values are indicated. (**C**) Microscopic images of Oil Red O-stained 3T3-L1 adipocytes after control and ZEB1 KD using three different quantities (measured in terms of µl) of pooled shRNAs at day 6. The relative expression of *Zeb1*, *Pparg* and *Adipoq* at day 2 are shown in the associated bargraphs.

In addition, we have listed the percentage of differentiated cells in response to control vector overexpression in [Supplementary-material SD1-data] of the revised manuscript*.* Finally, we added clarifying statements to the Materials and methods section of the revised manuscript.

As for the images in Figure 1, we note that these were not derived from the screen. Here, the differentiation difference results from the fact that panel 1C involves stable cell lines whereas panel 1D involves non-selected cells. We have consistently observed a lower differentiation capacity of stable cell lines compared to cells that are only transiently transduced. The differentiation enhancement (or lack thereof) needs therefore to be evaluated as compared to the stably or transiently transduced controls respectively. As suggested by the reviewers, we have added microscopic images of the cells shown in Figure 1 in two new figures: Figure 1—figure supplement 2 and Figure 1—figure supplement 3 of the revised manuscript and have indicated this in the revised Figure 1 legend.

Regarding the Western blot in Figure 1—figure supplement 1, several overexpressed TFs yielded indeed multiple HA-tagged products. This is especially the case for ZEB1. However, similar ZEB1 band profiles were also observed in other studies using different endogenous antibodies (Chang et al., Nat Cell Biol, 2001; Gregory et al., Nature Cell Biology, 2008; Wellner et al., Nat Cell Biology, 2009; Masin et al., *Cancer* and Metabolism, 2014). We therefore suspect that these bands reflect cryptic translation or specific protein degradation products given that they stem from the same open-reading frame construct and that they are all tagged by HA. To clarify this, we have added a statement to the revised Figure 1—figure supplement 1 legend.

2) Effect of ZEB1 knockdown on ZEB1 promoter occupancy. The ChIP-seq for Zeb1 should have a knockdown control. There are a large number of weak peaks, most of which seem not to have the ZEB1 motif. How many of these peaks are still present in the absence of ZEB1?

We already indicated in the original manuscript (Material and methods, Lentiviral production and stable cell lines) that despite extensive efforts, we were unable to derive a stable ZEB1 knockdown 3T3-L1 cell line. This issue renders the negative control experiment suggested by the reviewers rather difficult because of the intrinsic cellular heterogeneity that arises upon transient knockdown. We thus decided to validate our ZEB1 ChIP-seq data by performing a ChIP on HA-tagged ZEB1 in undifferentiated 3T3-L1 cells using a HA-specific antibody to provide the requested assessment of the specificity of the ZEB1 antibody. While overexpression may increase the number of binding events, we would expect the majority of endogenous ZEB1 binding sites to overlap with those detected using ZEB1-HA ChIP. Indeed, we found that ZEB1 replicate experiments versus ZEB1-HA before differentiation showed high correlation of read counts with Spearman’s rho = 0.83 and 0.86, in the range of what is typically observed for replicate ChIP-seq experiments (see revised Figure 3—figure supplement 1). A clarifying statement was added in the Results section of the revised manuscript).

In addition, we included in the revised manuscript the ChIP-qPCR results of 12 positive and three negative regions that we had previously used to validate the ZEB1 ChIP-seq libraries. Ten of these 12 positive regions were enriched compared to the negative regions further validating the robustness of the ZEB1 ChIP-seq data (revised Figure 3—figure supplement 1 and the respective qPCR primers were added to the revised Supplementary file 9).

*3) Relationship of ZEB1 and C/EBPb. The authors convincingly show that ZEB1 binds to many chromatin regions together with C/EBPβ and this factor also associates with ZEB1 in a co-IP experiment indicating that these factors cooperate on chromatin. This conclusion would be strengthened if it could be demonstrated that C/EBPβ binding to chromatin is affected by ZEB1 knockdown and vice versa*.

As indicated above, we were unable to derive stable ZEB1 knockdown cells, which makes performing ChIP experiments difficult. We therefore set out to analyze the binding behavior of ZEB1 in stable C/EBPβ knockdown 3T3-L1 cells (revised Figure 3—figure supplement 2). We selected 10 genomic regions bound by both C/EBPβ and ZEB1 as well as six regions bound by ZEB1 only, using publicly available C/EBPβ ChIP-seq data (Siersbaeck et al., EMBO J, 2011). First, we validated these regions by performing C/EBPβ ChIP-qPCR in control 3T3-L1 cells (revised Figure 3—figure supplement 2), allowing us to uncover that C/EBPβ binding is more prevalent than so far appreciated (65, 71) given that we found that most tested “ZEB1-only” regions (five out of six) are nevertheless bound by C/EBPβ. These results already strengthen the regulatory relationship between ZEB1 and C/EBPβ. We then targeted nine regions that we experimentally determined were bound by both C/EBPβ and ZEB1 as well as the sole ZEB1-only bound region to determine the ZEB1 DNA binding enrichment at these regions in stable C/EBPβ knockdown versus control cells using ZEB1 ChIP-qPCR. We found that ZEB1 DNA binding is decreased to various degrees at all tested regions compared to control knockdown cells, whereas this was not observed at the control site bound by ZEB1 only (revised Figure 3—figure supplement 2). These results suggest that the binding behavior of ZEB1 is influenced by C/EBPβ, although we cannot exclude at this point that the observed decrease in ZEB1 DNA binding in C/EBPβ knockdown cells may stem from a decrease in overall ZEB1 protein levels. Indeed, further analyses revealed that the knockdown of C/EBPβ in 3T3-L1 cells decreases ZEB1 gene expression, as assessed by qPCR (revised Figure 3—figure supplement 2). This result implies that C/EBPβ mediates *Zeb1* expression, which is substantiated by the fact that C/EBPβ appears to target the *Zeb1* gene in 3T3-L1 pre-adipocytes (revised Figure 3—figure supplement 2). Thus, the decreased DNA binding of ZEB1 in C/EBPβ-knockdown cells may also be a consequence of overall decreased ZEB1 levels (even though this was not reflected at the negative control site 10).

Taken together, although further experiments will be required to examine putative DNA binding cooperativity effects between ZEB1 and C/EBPβ, our data clearly demonstrate that ZEB1-C/EBPβ site co-occupancy is even more abundant than previously anticipated and that C/EBPβ may regulate ZEB1 expression, providing further support of the important regulatory relationship between ZEB1 and C/EBPβ. We have added these results to the revised manuscript as a new paragraph in the section “ZEB1 co-binds adipogenic regulatory regions with established early regulators such as C/EBPβ”**,** as well as in the revised Figure 3—figure supplement 2.

*4) Better correlation between gene expression and promoter occupancy. The correlation between ChIP-seq data and gene expression profiles is well documented (*Figure 4
*and supplemental information). However, it would be informative if the RNA-seq data upon ZEB1 knock down were included here to make it clear whether genes associated with early, late, or static ZEB1 binding are in fact affected by ZEB1 knock down. The data is already used in*
Figure 2*, so it should be straightforward to analyze the effect of knock down on the expression of putative direct ZEB1 target genes identified by ChIP-seq*.

As per the reviewers’ suggestion, we have now integrated the RNA-seq data upon ZEB1 knockdown with the dynamic ZEB1 binding data, specifically adding the text below to the section “ZEB1 DNA-binding is dynamic at adipogenic genes” of the revised manuscript as well as revised panels Figure 4 and Figure 4—figure supplement 1.

*5) Additional data for the in vivo studies. For the in vivo differentiation of implanted precursor cells, it would be relevant to know whether ZEB1 knockdown affects proliferation of precursor cells, their commitment or just their maturation to adipocytes. Also, data on gene expression would be informative*.

For the *in vivo* experiments, we were unfortunately unable to recover data on gene expression as the adipose tissues were fixed in paraformaldehyde and embedded with paraffin for H&E staining. However, as this specific method has been validated extensively (see e.g. Meissburger et al. EMBO Mol Med, 2011), we are confident that, consistent with our *in vitro* data, the effect observed here clearly demonstrates that ZEB1 is an important regulator of adipogenesis.

However, based on our image analyses, we were able to address the question whether ZEB1 modulation affects cell proliferation, which is a major determinant for adipogenesis. Specifically, we quantified the nuclei from the implant sections and did not observe any change in proliferation in the ZEB1 knockdown or overexpression samples compared to the respective controls, although ZEB1 knockdown tended to yield more variable results (revised Figure 6—figure supplement 1). These results suggest that the observed effect of ZEB1 on adipogenesis does not involve major changes in the degree of cell proliferation capacity. We have added this result to the “ZEB1 is an important regulator of in vivo adipogenesis” Results section of the revised manuscript.

Finally, we think that addressing the question whether commitment or maturation is affected is still beyond the current state of the art. This is mainly due to the fact that so far no bona fide marker exists to differentiate between cells that can commit to the adipocyte lineage versus committed adipoblasts capable of maturating into an adipocyte.

*6) Clarification of the human data. The human data in Figure 7 is somewhat at odds with prior results. Is adiponectin positively correlated with body weight in these samples? It looks like it might be, which is not generally how those two variables interact*.

While we acknowledge that the correlations presented here might suggest a positive correlation between adiponectin and body weight, this is in fact not the case (Figure 7). It has to be mentioned that the correlation of adiponectin with body weight is seen when comparing lean versus obese individuals. The cohort that we employed in this study consists entirely of obese individuals with a BMI of >35. Thus, no correlation between adiponectin and body weight was necessarily expected and none was indeed observed.

7) Claims regarding commitment are premature. A weak point in the manuscript is the data on the importance of ZEB1 for commitment of MSCs to the adipocyte lineage. Although the authors provide evidence that ZEB1 is required for terminal differentiation of MSCs (i.e. C3H10T1/2 cells), it is not clear if this is because ZEB1 is required for the commitment step. The analysis of expression of Zfp423 and Zfp521 upon ZEB1 knock down is not convincing in itself. If the authors want to claim that ZEB1 also regulates commitment, this should be more thoroughly evaluated experimentally. Note, the editor believes that this question is better suited for a future manuscript and strongly suggests that the authors simply soften their conclusions regarding a role in commitment in the present paper. Alternatively, if the authors wish to make this claim, additional studies would be required. Some of the questions that could be addressed include: (1) can over expression of ZEB1 substitute for BMP-2/4 signaling to commit C3H10T1/2 cells to the adipocyte lineage?; (2) does overexpression of ZEB1 decrease the potential of C3H10T1/2 cells to develop into other cell lineages?; (3) is transient overexpression or knock down during commitment enough to enhance or decrease, respectively, differentiation of C3H10T1/2 cells?; or (4) is overexpression or knock down only during terminal differentiation of C3H10T1/2 enough to enhance or inhibit differentiation, respectively?

We acknowledge that our current C3H10T1/2 experiments, while highly suggestive, may not be sufficient to conclusively demonstrate ZEB1’s involvement in adipogenic commitment. We therefore have decided to take the editor’s advice and to pursue this question in a follow-up study. In the current manuscript, we now present ZEB1’s intriguing, positive regulatory effect on adipogenic commitment factors and on C3H10T1/2 adipogenesis in the context of the adipogenic regulatory network. To this end, we have restructured the last two figures, in what we think is a concise and conceptually cohesive way of presenting the mesenchymal stem cell results. Additionally, we have removed “determination” from the Abstract and it now reads “identifying ZEB1 as a central transcriptional component of fat cell differentiation”.